# AVGen-Bench: A Task-Driven Benchmark for Multi-Granular Evaluation of Text-to-Audio-Video Generation

Ziwei Zhou [* 1]   Zeyuan Lai [* 2]   Rui Wang [1]   Yifan Yang [3]   Yuqing Yang [3]   Qi Dai [3]   Lili Qiu [3]   Chong Luo [3]

## Abstract

Text-to-Audio-Video (T2AV) generation is rapidly becoming a core interface for media creation, yet its evaluation remains fragmented. Existing benchmarks largely assess audio and video in isolation or rely on coarse embedding similarity, failing to capture fine-grained joint correctness required by realistic prompts. We introduce **AVGen-Bench**, a task-driven benchmark for T2AV generation, featuring high-quality prompts across 11 real-world categories. To support comprehensive assessment, we propose a multi-granular evaluation framework that combines lightweight specialist models with Multimodal Large Language Models (MLLMs), enabling evaluation from perceptual quality to fine-grained semantic controllability. Our evaluation reveals a pronounced gap between strong audio-visual aesthetics and weak semantic reliability, including persistent failures in text rendering, speech coherence, physical reasoning, and universal breakdown in musical pitch control. Code and benchmark resources are available at http://aka.ms/avgenbench.

## 1. Introduction

The landscape of generative video is undergoing a fundamental shift from *silent* Text-to-Video (T2V) synthesis (OpenAI, 2024; Wan et al., 2025; Wu et al., 2025) to multimodal Text-to-Audio-Video (T2AV) generation (Low et al., 2025; HaCohen et al., 2026; AI, 2026). This transition is not merely an incremental feature upgrade. In many real-world AIGC scenarios, audio is essential for conveying information, realism, and engagement. A visually plausible video

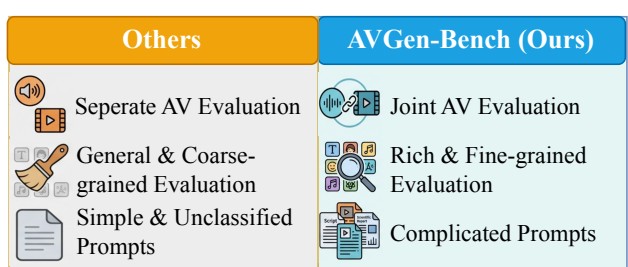

*Figure 1.* **Comparison between AVGen-Bench and existing benchmarks.** Unlike prior works that rely on separate audio/visual evaluations and simple prompts, AVGen-Bench introduces (1) joint audio-visual evaluation, (2) fine-grained metrics across 10 dimensions, and (3) rich, complex prompts with high token counts to ensure a rigorous assessment.

without sound is often flat and uninformative, while synchronized and semantically correct audio can dramatically enhance immersion—for example, the crisp cutting sound in a fruit-slicing clip or intelligible dialogue in a conversational scene. As frontier systems such as Sora 2 (OpenAI, 2025), Veo 3.1 (DeepMind, 2026b), and Kling 2.6 (KuaishouTechnology, 2026) emerge, T2AV generation is quickly becoming the default interface for user-centric creation.

Despite rapid architectural progress, the field faces a critical bottleneck: the lack of a rigorous and holistic evaluation framework for T2AV. Most existing benchmarks for generative models were designed for *uni-modal* settings. Visual benchmarks such as VBench (Huang et al., 2024) and VBench++ (Huang et al., 2025) focus exclusively on video quality, while audio benchmarks typically evaluate sound in isolation. More recent efforts attempt to combine audio and video evaluation (Wang et al., 2025a; Zhang et al., 2025; Liu et al., 2025; Hu et al., 2025), but they still fall short in two key aspects. First, they often rely on coarse-grained metrics that score overall audio, video, or audio-visual quality, without distinguishing specific capabilities or failure modes. Second, joint evaluation is commonly reduced to embedding similarity using models such as CLIP (Radford et al., 2021) or CLAP (Wu et al., 2023), which is insufficient for verifying fine-grained semantic alignment required by realistic prompts.

This limitation becomes particularly evident in real T2AV usage. Users typically provide a single textual prompt that

---

[*]Equal contribution  [1]Fudan University  [2]University of Science and Technology of China  [3]Microsoft Research Asia. Correspondence to: Yifan Yang <yifanyang@microsoft.com>.

*Proceedings of the $43^{rd}$ International Conference on Machine Learning*, Seoul, South Korea. PMLR 306, 2026. Copyright 2026 by the author(s).

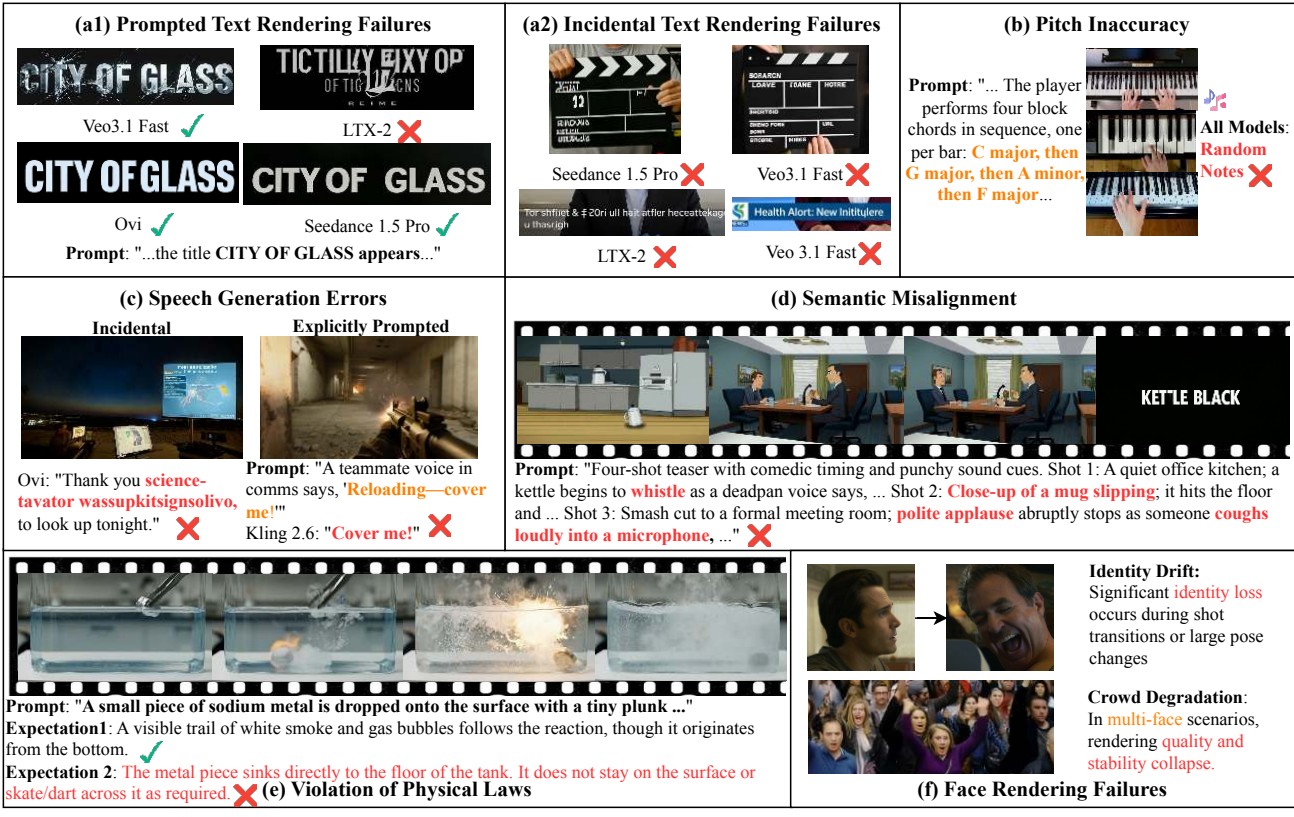

*Figure 2.* **Qualitative examples of failure modes across different fine-grained dimensions. (a1)** Explicitly prompted text rendering. **(a2)** Incidental text rendering in background elements. **(b)** Fine-grained musical control (Pitch Accuracy). **(c)** Speech generation regarding incidental coherence and explicit instruction following. **(d)** Holistic semantic alignment in complex multi-shot narratives. **(e)** High-level physical plausibility and dynamic constraints. **(f)** Facial consistency failures, illustrating identity drift across shot transitions and degradation in multi-face crowd scenes. Red markings and crosses (✗) indicate generated errors.

interleaves visual and acoustic requirements—often implicitly—rather than specifying audio and video separately. Under such settings, current models exhibit recurring yet under-measured failure modes: speech content that is unintelligible or incorrect, environmental sounds that do not align with visual events, mismatched lip movements, incorrect musical notes despite realistic playing motions, and violations of basic physical or causal logic. Figure 2 illustrates representative examples of these phenomena. Without a benchmark that explicitly targets these joint, fine-grained behaviors, it is difficult to diagnose model weaknesses or guide future progress.

To bridge this gap, we introduce **AVGen-Bench**, a task-driven benchmark dedicated to Text-to-Audio-Video generation. Instead of tailoring prompts to fit available metrics, AVGen-Bench is grounded in realistic user intents and application scenarios. Our prompt suite spans 11 daily-life categories, covering professional media production (e.g., movie trailers and advertisements), creator economy applications (e.g., music tutorials and gameplay), and physically grounded world simulation tasks. This task-centric design

enables meaningful evaluation of not only perceptual quality, but also whether a model can *accomplish what the user intends* in a given scenario.

Furthermore, we propose a comprehensive, multi-granular evaluation suite for T2AV. Beyond basic uni-modal aesthetics and audio-visual synchronization, our framework introduces targeted metrics for fine-grained controllability and semantic correctness, including scene text legibility, facial identity consistency, pitch accuracy in music generation, speech intelligibility, and physical plausibility. Methodologically, we adopt a hybrid evaluation strategy that integrates lightweight specialist models with Multimodal Large Language Models (MLLMs). This design leverages the complementary strengths of both paradigms: specialist models provide precise signal-level measurements, while MLLMs enable high-level semantic reasoning and holistic intent verification.

In summary, our contributions are threefold: **(1) A Task-Driven T2AV Benchmark.** We present **AVGen-Bench**, a curated benchmark with high-quality prompts across 11 real-world categories, shifting evaluation from metric-driven

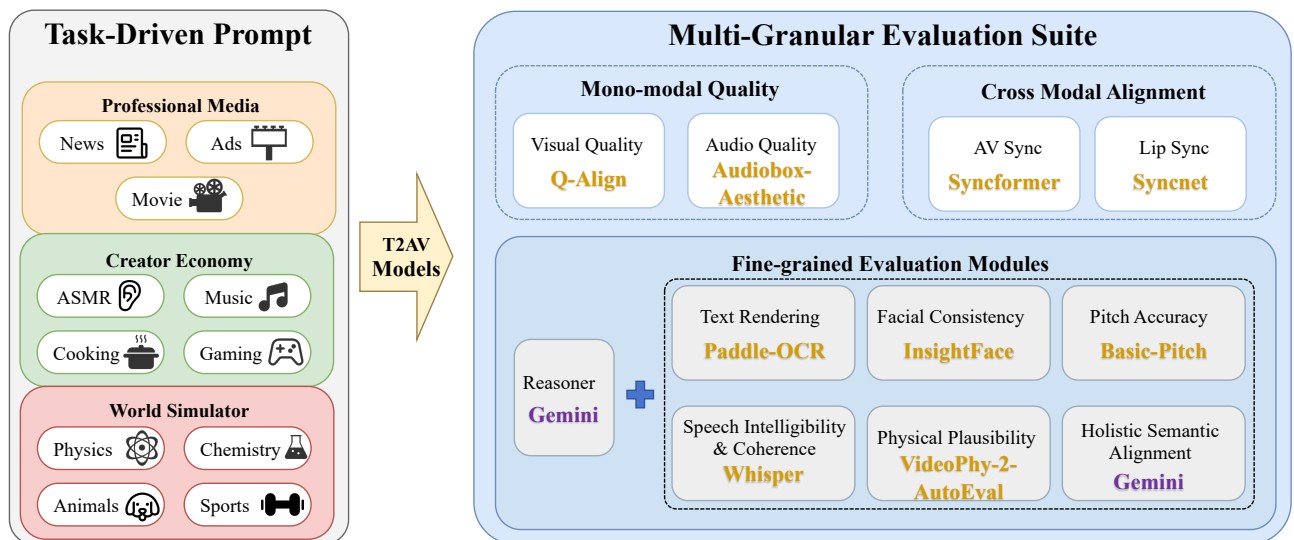

*Figure 3.* **Overview of the AVGen-Bench framework.** The benchmark features a **Task-Driven Prompt Set** (left) categorized into three real-world application domains: Professional Media, Creator Economy, and World Simulation. The generated content is evaluated via our **Multi-Granular Evaluation Suite** (right), which employs a hybrid strategy combining lightweight specialist models (orange) for signal-level precision and MLLMs (purple) for high-level semantic reasoning and physical plausibility analysis.

design to user-centric task understanding. **(2) A Multi-Granular, Hybrid Evaluation Framework.** We introduce a unified evaluation suite that jointly assesses uni-modal quality, audio-visual consistency, and fine-grained semantic alignment by combining specialist models with MLLMs. **(3) A Systematic Diagnosis of T2AV Failure Modes.** Through extensive evaluation, we reveal a sharp gap between strong audio-visual aesthetics and weak fine-grained semantic control, highlighting critical challenges in text, speech, and physical reasoning.

**Conflict of Interest Disclosure.** Several authors are affiliated with Microsoft Research Asia. In this work, Sora 2 was accessed through an Azure-hosted deployment, while other commercial and open-source systems were evaluated through their respective standard APIs or official checkpoints. We disclose this affiliation and infrastructure access for transparency. All evaluated systems were tested using the same benchmark protocol, and the authors' institutional affiliations did not influence the evaluation procedure or reported results.

## 2. Related Works

### 2.1. Audio-Video Generation Models

**Text-to-Video (T2V) Synthesis.** The advent of Sora (OpenAI, 2024) marked a paradigm shift, demonstrating the scalability of Diffusion Transformers (DiT) (Peebles & Xie, 2023) for video synthesis. This catalyzed a rapid transition from earlier U-Net architectures(Blattmann et al., 2023; Guo et al., 2024) to DiT and Flow Matching (Lipman et al.,

2023) paradigms. Consequently, a wave of high-fidelity T2V models has emerged, ranging from proprietary systems (KuaishouTechnology, 2026; Runway, 2024) to powerful open-weight ones such as HunyuanVideo (Wu et al., 2025), LTX-Video (HaCohen et al., 2024) and Wan (Wan et al., 2025). Despite achieving cinema-grade visual quality, these "silent" models lack the acoustic dimension essential for immersive world modeling.

**Joint Audio-Video Generation.** To bridge the modality gap, research has pivoted towards unified T2AV architectures. Leading proprietary systems, including **Sora 2** (OpenAI, 2025), **Veo 3.1** (DeepMind, 2026b), **Wan 2.6** (AI, 2026), and **Kling 2.6** (KuaishouTechnology, 2026), demonstrate high-fidelity synchronized synthesis. In the open domain, models like **Ovi** (Low et al., 2025) and **JavisDiT** (Liu et al., 2025) explore dual-stream Diffusion Transformers, while **LTX-2** (HaCohen et al., 2026) employs flow matching. Recently, hybrid architectures such as **MAViD** (Pang et al., 2025) have emerged, combining Autoregressive (AR) modeling with diffusion to enhance cross-modal consistency. Complementary to these unified approaches are conditional pipelines, including Video-to-Audio (e.g., **MMAudio** (Cheng et al., 2025), **Kling-Foley** (Wang et al., 2025c)) and Audio-to-Video (e.g., **MTVCraft** (Weng et al., 2025), **Wan-S2V** (Gao et al., 2025)) systems. While useful, these cascaded methods differ fundamentally from the holistic world modeling aim of simultaneous generation.

### 2.2. Text-to-Audio-Video Benchmarks

Prior protocols typically isolate modalities. In the visual domain, the **VBench** series (Huang et al., 2024; 2025; Zheng

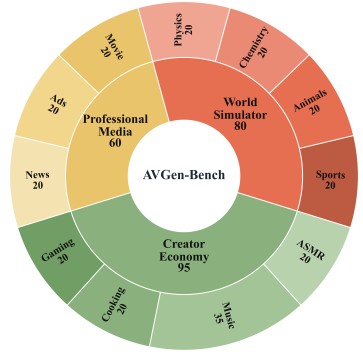

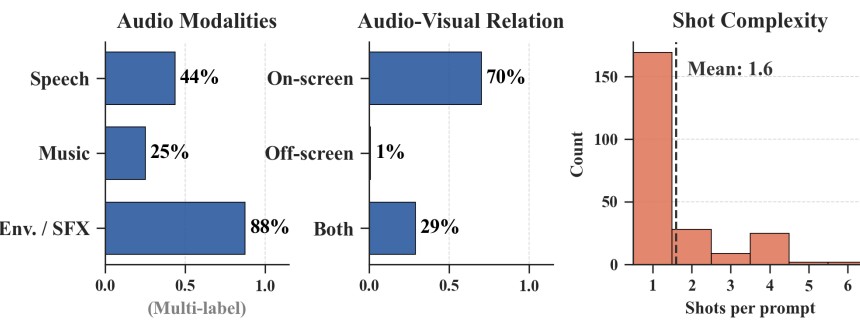

*(a)* Distribution of the 235 curated prompts across 3 main domains and 11 sub-categories.

*(b)* Distribution of audio types, audio source relation, and shot counts.

*Figure 4.* **Dataset-level statistics of the prompts used in AVGen-Bench.**

*Table 1.* **Comparison with existing benchmarks.** AVGen-Bench features the highest average prompt complexity (Avg. Tokens) and a comprehensive set of evaluation metrics covering all audio modalities.

| Benchmark | Task | #Metrics | Avg. Tokens | Audio Types |
|---|---|---|---|---|
| VBench 2.0 | T2V | 18 | 26.56 | - |
| TTA-Bench | T2A | 10 | 20.00 | SFX, Music, Speech |
| JavisBench | T2AV | 4 | 65.00 | SFX, |
| Harmony-Bench | TI2AV | 6 | - | SFX, Music, Speech |
| VerseBench | TI2AV | 4 | 68.00 | SFX, Speech |
| UniAVGen | TI2AV | 3 | - | Speech |
| **AVGen-Bench (Ours)** | **T2AV** | **10** | **88.54** | **SFX, Music, Speech** |

et al., 2025) sets the standard for video quality but inherently neglects the acoustic dimension. Conversely, audio benchmarks like **TTA-Bench** (Wang et al., 2025b) focus on text-to-audio generation but often face scalability bottlenecks, relying heavily on subjective human evaluation to compensate for the poor perceptual correlation of traditional automated metrics. Recent studies have attempted to combine audio and video evaluation into unified benchmarks. Early works such as **HarmonyBench** (Hu et al., 2025), **UniAVGen** (Zhang et al., 2025), and **VerseBench** (Wang et al., 2025a) assess the capability to generate both modalities together. However, these benchmarks are often too general (coarse-grained). They typically score overall audio-visual quality but fail to distinguish specific errors, such as incorrect pitch or rhythm.

Similarly, benchmarks like **JavisBench**(Liu et al., 2025) and **TAVGBench** (Mao et al., 2024) rely heavily on distribution- or embedding-based metrics such as FVD, FAD, ImageBind, CLIP(Radford et al., 2021), and CLAP (Wu et al., 2023). While useful for general fidelity and coarse matching, these metrics cannot verify fine-grained details like specific musical notes, precise text legibility, or strict synchronization. **MTAVG-Bench** (Zhou et al., 2026) provides an important diagnostic framework for multi-talker dialogue-centric generation, but its scope naturally excludes broader daily-life scenarios such as non-speech sound design, object interac-

tions, chemical reactions, and acoustic resonance. More recent broad benchmarks, including **VABench** (Hua et al., 2025) and **T2AV-Compass** (Cao et al., 2025), are closer to our setting. In contrast, AVGen-Bench focuses on more targeted, failure-prone dimensions such as strict musical pitch accuracy, multi-shot facial consistency, physical plausibility, and holistic semantic alignment, and uses specialist-model parsing plus MLLM reasoning rather than relying solely on direct MLLM judgment. We provide a comparison between our benchmark and existing benchmarks in Table 1.

## 3. AVGen-Bench

This section details the architecture of **AVGen-Bench**. We begin by outlining our task-driven prompt construction strategy, structured around diverse daily-life categories to probe model capabilities boundaries. Following this, we introduce our evaluation protocol, discussing the rationale behind our hybrid design and specifying the implementation of individual metrics for uni-modal quality, cross-modal alignment, and fine-grained semantic control.

### 3.1. Task-Driven Prompt Curation

To ensure our benchmark reflects realistic usage rather than merely categorizing static visual concepts, we adopt a top-down, intent-first curation strategy. We first defined a comprehensive taxonomy of user scenarios for AI video generation, and then implemented a "Human-in-the-Loop" generation pipeline. Specifically, we utilized GPT-5.2 (OpenAI, 2026) to generate candidate prompts based on our scenario definitions, followed by a rigorous manual review process to filter for complexity, clarity, and diversity.

As illustrated in Figure 4a, the resulting dataset consists of **235** highly curated tasks, systematically distributed across 3 main domains and 11 real-world sub-categories. Notably, to simulate professional editing workflows, the dataset main-

tains an average of 1.6 shots per prompt, with 44% of samples involving speech and 88% containing environmental sound effects as demonstrated in Figure 4b.

Crucially, a distinct feature of our framework is that the prompt curation is entirely decoupled from the evaluation metrics. Unlike prior benchmarks that often reverse-engineer prompts to fit specific available detectors (e.g., curating speech prompts solely because a TTS metric is available), our prompts are derived strictly from genuine user needs. This design choice ensures that **AVGen-Bench** is both scalable and customizable—users can easily extend the prompt set to new domains. The resulting prompt set is organized into three task domains:

**Professional Media Production.** This domain assesses the model's capacity to synthesize cinema-grade content suitable for professional workflows. For *Commercial Ads*, we curated a dataset of classic Bumper Ads from YouTube and employed Gemini 3 Pro (DeepMind, 2026a) to reverse-caption these videos into anonymized textual descriptions, ensuring the prompts describe visual styles without relying on specific brand logos. For *Movie Trailers*, we instructed GPT-5.2 to construct multi-shot scripts, requiring the model to maintain visual consistency and narrative continuity across varying camera angles and scene transitions.

**Creator Economy.** Geared towards the booming sector of user-generated content, this domain covers ASMR, Cooking Tutorials, Gameplays, and Musical Instrument Tutorials. A critical innovation in the *Musical Instrument Tutorial* category is the injection of fine-grained acoustic constraints. We explicitly included requirements for specific musical scales (e.g., "C Major scale") or chords in the prompts. This design rigorously tests whether the model can perform precise audio-visual alignment—generating the correct audio frequencies corresponding to the visual finger positions—rather than merely producing generic music.

**World Simulator.** This domain probes the model's understanding of fundamental laws governing the physical world, spanning Physics, Chemistry, Sports, and Animals. Notably, for *Physics* and *Chemistry*, we employed an "**Underspecified Prompting**" strategy. In these prompts, we intentionally omit explicit descriptions of the physical outcome. For example, in a prompt describing a Newton's Cradle experiment, we describe the setup but do not specify how many balls should recoil. This forces the model to rely on its "world knowledge" to simulate the correct physical dynamics, rather than simply following a textual instruction.

### 3.2. Evaluation Suite

To provide a holistic assessment of generative quality, we construct a comprehensive evaluation suite for **AVGen-Bench** that utilizes a hybrid methodology, integrating lightweight specialist models with Multimodal Large Language Models (MLLMs). This architecture allows us to bridge the gap between low-level signal fidelity and high-level semantic reasoning, covering three critical dimensions: uni-modal aesthetics, cross-modal alignment, and text-to-media consistency. Furthermore, we introduce a set of targeted evaluation modules specifically designed to probe capabilities where current models empirically struggle, such as scene text rendering and fine-grained audio control.

#### 3.2.1. BASIC EVALUATION MODULES

**Uni-modal Quality.** We begin by assessing the perceptual quality of the visual and acoustic modalities independently. For the visual domain, we leverage **Q-Align** (Wu et al., 2024), a state-of-the-art MLLM-based evaluator fine-tuned to correlate closely with human aesthetic judgments. Unlike distribution-based metrics (e.g., FVD (Unterthiner et al., 2019)), Q-Align provides a direct score reflecting visual fidelity and technical quality. For the audio domain, we utilize the aesthetic assessment module from **Audiobox** (Tjandra et al., 2025) (*Audiobox-Aesthetic*). This model evaluates acoustic clarity and production quality, serving as a robust proxy for subjective listening tests.

**Cross-Modal Alignment.** Proper timing between video and audio is crucial for creating realistic content. We evaluate this using two specific methods. For general synchronization (e.g., impact sounds), we use **Syncformer** (Iashin et al., 2024). It calculates the time difference between visual motion and the start of the sound. Additionally, since humans are very sensitive to mismatched speech, we use the standard **SyncNet** (Chung & Zisserman, 2016) model for *Lip Synchronization*. This measures the error (in frames) between lip movements and speech, ensuring that characters appear to speak naturally.

#### 3.2.2. FINE-GRAINED EVALUATION MODULES

General aesthetic metrics often gloss over specific semantic failures. To address this, we introduce a suite of **hybrid evaluation pipelines**. By chaining specialist models (as feature extractors) with Gemini 3 Flash (as the reasoning engine), we can rigorously audit the model's adherence to fine-grained constraints.

**Scene Text Rendering.** To evaluate the accuracy and contextual validity of generated text, we implement a "detect-aggregate-verify" pipeline. First, we utilize **PaddleOCR** (Cui et al., 2025a) to extract text content and bounding boxes from each video frame. Addressing temporal redundancy, we apply a spatiotemporal clustering algorithm to aggregate spatially proximal text instances across adjacent frames into consolidated sequences. Finally, these parsed sequences are fed into the MLLM for a **dual-objective assessment**: (1) verifying strict adherence to any text explicitly specified in

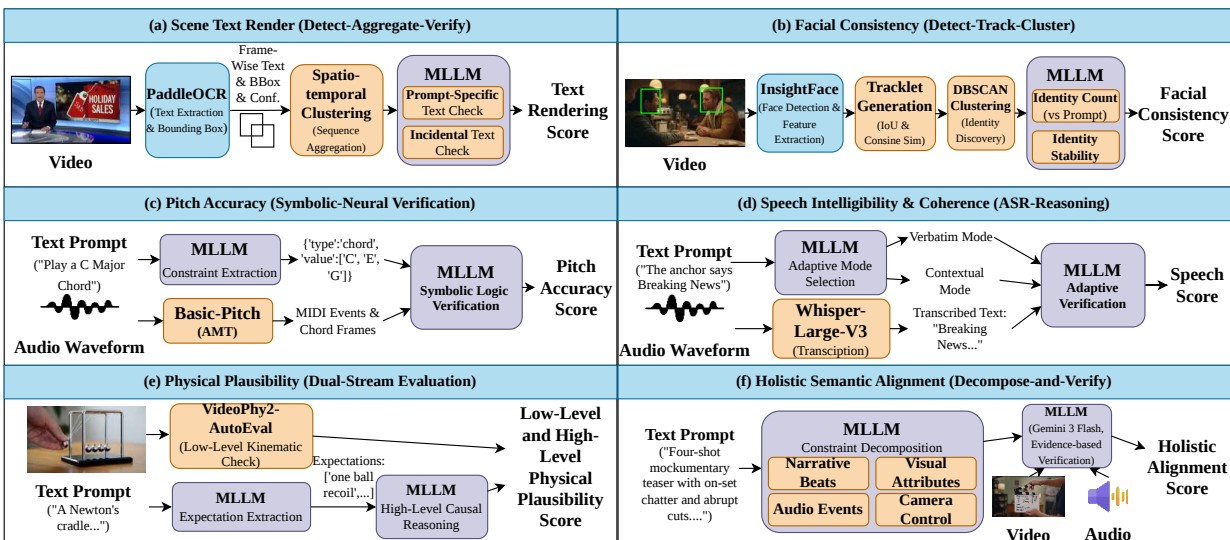

*Figure 5.* **Detailed workflows of the six Fine-grained Evaluation Modules in AVGen-Bench.** The suite employs hybrid strategies combining specialist models (blue nodes) and MLLMs (purple nodes) to evaluate: **(a)** Scene Text Rendering (OCR + Verification); **(b)** Facial Consistency (InsightFace + DBSCAN); **(c)** Pitch Accuracy (Audio-to-MIDI + Theory Check); **(d)** Speech Intelligibility (ASR + Contextual Logic); **(e)** Physical Plausibility (Kinematics + Causal Reasoning); and **(f)** Holistic Semantic Alignment (Constraint Decomposition).

the prompt, and (2) evaluating the **semantic coherence of incidental text** (e.g., scrolling tickers in news broadcasts). This ensures that even unprompted text elements are legible and contextually appropriate, rather than manifesting as gibberish or visual artifacts.

**Facial Consistency.** To quantify identity preservation and stability without referencing external character facial features, we implement a reference-free "Detect-Track-Cluster" pipeline augmented by MLLM-derived constraints. We first employ **InsightFace** (Buffalo-L) (Deng et al., 2019) to extract facial embeddings and bounding boxes frame-by-frame. To handle occlusion and temporal discontinuity, we construct "tracklets" using a hybrid heuristic combining IoU overlap and cosine similarity. Subsequently, we apply **DBSCAN** clustering on these tracklets to discover distinct identities (clusters), filtering for "primary characters" based on temporal occupancy ratios. The final consistency score is a weighted aggregate of two dimensions: **(1) Identity Count Accuracy (40%):** We compare the number of discovered primary clusters against the ground-truth character count predicted by **Gemini** based on the prompt, penalizing hallucinations or erasure. **(2) Identity Stability (60%):** For each primary cluster, we measure the 50th percentile ($P_{50}$) internal cosine similarity of its tracklets to assess the robustness of identity preservation over time.

**Pitch Accuracy.** General audio encoders fail to verify fine-grained music theory constraints. We address this via a Symbolic-Neural Verification pipeline. First, we feed the text prompt into **Gemini** to perform *Constraint Extraction & Gating*, extracting explicit musical constraints (e.g., "C

Major chord") into a structured JSON format while filtering out abstract prompts (e.g., "jazzy vibe") to avoid invalid penalization. For applicable prompts, we then employ **Basic-Pitch** (Bittner et al., 2022) for *Automatic Music Transcription (AMT)*, converting the audio waveform into symbolic MIDI events and aggregating note onsets within an 80ms window into "chord frames." Finally, the extracted MIDI events are fed back to Gemini for *Symbolic Logic Verification*, where the MLLM verifies whether the generated note sequences strictly adhere to the music theory requirements defined in the prompt.

**Speech Intelligibility & Coherence.** Unlike general audio metrics, we aim to verify the semantic content of speech using a cascade ASR-Reasoning pipeline. We utilize **Faster-Whisper** (Radford et al., 2022), which integrates Voice Activity Detection (VAD) to effectively filter non-speech noise and accelerate inference, for robust transcription. We then employ **Gemini** for semantic auditing, introducing an Adaptive Compliance Mechanism. Specifically, in *Verbatim Mode* (triggered when the prompt explicitly prescribes dialogue), the pipeline enforces strict lexical matching. Conversely, in *Contextual Mode* (for prompts implying speech without specifying content), the system evaluates *Semantic Coherence*, detecting whether the generated speech aligns with the visual context and narrative intent or degenerates into unintelligible gibberish.

**Physical Plausibility.** We evaluate physical realism through two decoupled modules targeting different levels of abstraction. For *Low-Level Kinematic Plausibility*, we employ **VideoPhy2-AutoEval** (Bansal et al., 2025). This special-

ist model acts as a "physics engine checker," scoring the video based on motion smoothness and trajectory realism to detect basic artifacts like jittery motion independent of semantic context. In parallel, for *High-Level Causal Reasoning* (e.g., "Sodium dropped into water"), we implement a Two-Stage Semantic Verification pipeline using **Gemini** inspired by PhyT2V (Xue et al., 2025). This involves first extracting a list of **Observable Expectations** (e.g., "violent bubbling") from the prompt, followed by **Semantic Adjudication**, where the MLLM logs observable events in the video to calculate a **Semantic Physics Score** based purely on the alignment between expected physical outcomes and visual evidence.

**Holistic Semantic Alignment.** While embedding-based metrics capture high-level relevance, they often fail to penalize subtle contradictions. To address this, we implement a "Decompose-and-Verify" pipeline using **Gemini** as a multimodal auditor. The MLLM first performs *Constraint Decomposition*, parsing the prompt into checkable constraints across four dimensions: (1) **Narrative Beats**, (2) **Visual Attributes** (object counts, colors), (3) **Audio Events**, and (4) **Cinematography**. Subsequently, it performs *Evidence-based Scoring* by scanning the video to verify each constraint against visual/audio evidence. The final score provides a nuanced assessment of how well the generated content fulfills the user's intent beyond simple semantic similarity.

# 4. Experiment

## 4.1. Experimental Setup

**Models Evaluated.** To ensure a comprehensive assessment of the current T2AV landscape, we select a diverse set of state-of-the-art models spanning both commercial services and research frameworks. For proprietary systems, we evaluate market-leading models accessed via their official APIs, including **Sora 2** (OpenAI, 2025), **Kling 2.6** (KuaishouTechnology, 2026), **Wan 2.6** (AI, 2026), **Seedance-1.5 Pro** (Seedance et al., 2025), and **Seedance 2.0** (Seedance et al., 2026). Additionally, we include Google's **Veo 3.1** (DeepMind, 2026b), testing both its *Fast* and *Quality* variants to analyze the trade-off between inference speed and generation fidelity. In the open-source domain, we evaluate representative unified models, specifically **LTX-2.3**, **LTX-2** (HaCohen et al., 2026), and **Ovi** (Low et al., 2025). Furthermore, to benchmark modular cascaded approaches, we include a standard T2V+V2A pipeline combining **Wan 2.2** (Wan et al., 2025) with **HunyuanVideo-Foley** (Shan et al., 2025). We also incorporate Text-to-Image-to-Audio-Video (T2Image+TI2AV) pipelines by pairing both the open-source **Emu3.5** (Cui et al., 2025b) and the proprietary **NanoBanana2** (Raisinghani, 2026) with the open-source **MOVA** (SII-OpenMOSS Team et al., 2026)

model.

**Implementation Details.** To maintain a fair comparison, we standardize the output resolution for the majority of models to **720p** (1280×720), with the exception of pipelines utilizing **MOVA**, which are evaluated using its **360p** version. Regarding temporal duration, we target a length of **10 seconds** for most models (e.g., Kling 2.6, Wan 2.6, LTX-2.3, LTX-2, Ovi). Exceptions are dictated by specific architectural or API constraints: **Veo 3.1** is evaluated at 8 seconds (its maximum supported duration), **Sora 2** at **12 seconds** due to fixed duration quantization, and the **Wan 2.2** pipeline at **5 seconds** (16 fps) reflecting the T2V model's native generation limit. For open-source models, inference is performed using official checkpoints with default sampling parameters recommended by their respective authors.

## 4.2. Experimental Results

We provide detailed analysis of each evaluation module. To complement the quantitative metrics, we visually analyze representative failure cases across different fine-grained dimensions in **Figure 2**. Additional examples and extended analysis are provided in **Appendix B**. For compact model-level comparison, we also report a Total score in Table 2: Total $= 0.2S_{\text{basic}} + 0.2S_{\text{cross}} + 0.6S_{\text{fine}}$, where $S_{\text{basic}} = \text{mean}(\text{Vis} \times 100, \text{Aud(PQ)} \times 10)$, $S_{\text{cross}} = \text{mean}(100 \cdot \max(0, 1 - \text{AV}/0.5), 100 \cdot \max(0, 1 - \text{Lip}/8))$, and $S_{\text{fine}} = \text{mean}(\text{Text}, \text{Face}, \text{Music}, \text{Speech}, \text{Lo-Phy} \times 20, \text{Hi-Phy}, \text{Holistic})$.

**Basic Uni-modal Quality.** As presented in Table 2, the evaluated models demonstrate exceptional performance in the visual domain. The consistently high Visual Quality scores (e.g., Seedance-1.5 Pro reaching 0.970 and Veo 3.1 reaching 0.960) indicate that current T2AV systems have largely mastered the synthesis of high-fidelity imagery. Qualitative inspection confirms that this metric aligns strongly with subjective perception: models with top-tier scores consistently produce videos with professional lighting, composition, and "cinematic" aesthetics.

In contrast, Audio Quality scores—specifically measured by the **Production Quality (PQ)** sub-metric of Audiobox-Aesthetic—are relatively lower, suggesting that acoustic synthesis still trails behind visual generation. We observe a clear correlation between PQ and auditory clarity: high-scoring models (e.g., Seedance-1.5 Pro at 7.48) generate crisp, studio-like sound, whereas lower scores typically correspond to audible background noise or signal artifacts.

**Basic Cross-modal Alignment.** Regarding temporal synchronization, results indicate that current models have not yet achieved frame-perfect alignment. For general AV Sync, the mean absolute offset ranges from 0.2s to 0.44s, while Lip Sync errors span from 2.0 to over 5 frames. These

*Table 2.* **Quantitative comparison on AVGen-Bench.** We evaluate models across three granularities: Basic Uni-modal (Visual/Audio Aesthetic), Basic Cross-modal (Sync), and our proposed Fine-grained Modules. Best scores are highlighted in **bold**, and second-best are underlined. Note that for AV-Sync and Lip-Sync, lower (↓) is better; for others, higher (↑) is better. We also report an aggregate Total score (Scheme-2). Wan2.2+HunyuanVideo-Foley denotes a cascaded pipeline of T2V followed by V2A. Emu3.5+MOVA and NanoBanana2+MOVA are both T2Image+TI2AV cascaded pipelines. Proprietary components are marked with orange background, while open-source components are marked with blue background. Models are sorted by Overall score in descending order.

| Model | Basic Uni-modal | | Basic Cross-modal | | Fine-grained Visual | | Fine-grained Audio | | Fine-grained Macro | | | Overall |
|---|---|---|---|---|---|---|---|---|---|---|---|---|
| | Vis ↑ | Aud (PQ) ↑ | AV ↓ | Lip ↓ | Text ↑ | Face ↑ | Music ↑ | Speech ↑ | Lo-Phy ↑ | Hi-Phy ↑ | Holistic ↑ | Total ↑ |
| Seedance 2.0 | 0.945 | 7.15 | **0.15** | 4.14 | 74.83 | **60.95** | **28.12** | 94.09 | 3.89 | **83.16** | 89.61 | **72.07** |
| Veo 3.1-fast | 0.960 | 6.64 | 0.21 | 2.39 | 75.10 | 52.77 | 3.13 | 94.53 | 3.68 | 67.43 | 86.27 | 67.87 |
| Veo 3.1-quality | 0.954 | 6.77 | 0.24 | 3.59 | 76.53 | 52.90 | 5.00 | **96.09** | 3.74 | 68.53 | 84.10 | 66.28 |
| Sora-2 | 0.848 | 5.91 | 0.25 | 4.50 | 74.84 | 51.17 | 7.81 | 88.63 | **4.05** | 78.95 | 88.89 | 64.16 |
| Wan2.6 | 0.959 | 7.15 | 0.30 | 4.32 | **76.95** | 49.27 | 1.75 | 89.33 | 3.69 | 66.92 | 80.98 | 62.97 |
| Seedance-1.5 Pro | **0.970** | **7.48** | 0.26 | 3.43 | 38.28 | 54.42 | 1.88 | 93.45 | 3.72 | 66.88 | 77.38 | 62.55 |
| Kling-V2.6 | 0.906 | 6.93 | 0.21 | 2.30 | 14.52 | 57.33 | 5.00 | 89.62 | 3.84 | 63.92 | 76.74 | 61.82 |
| LTX-2.3 | 0.858 | 7.11 | 0.36 | **2.00** | 54.17 | 45.06 | 1.38 | 86.66 | 3.99 | 64.31 | 65.22 | 59.97 |
| NanoBanana2 + MOVA | 0.890 | 6.71 | 0.44 | 2.70 | 68.26 | 41.33 | 0.59 | 82.45 | 3.91 | 60.95 | 72.48 | 58.10 |
| LTX-2 | 0.828 | 6.84 | 0.23 | 4.76 | 24.76 | 48.53 | 5.75 | 87.07 | **4.05** | 60.20 | 66.59 | 56.62 |
| Emu3.5 + MOVA | 0.911 | 6.80 | 0.38 | 4.83 | 64.72 | 48.44 | 0.62 | 81.74 | 3.89 | 55.85 | 66.55 | 56.12 |
| Wan2.2 + HunyuanVideo-Foley | 0.936 | 6.60 | 0.23 | 5.38 | 48.46 | 36.23 | 3.44 | 53.40 | 3.90 | 54.11 | 60.63 | 53.29 |
| Ovi | 0.839 | 6.31 | 0.37 | 5.40 | 41.36 | 49.05 | 11.25 | 76.49 | 3.93 | 52.92 | 57.45 | 52.02 |

figures reveal a tangible gap from ideal performance, particularly in speech scenarios where even minor offsets (e.g., > 2 frames) can disrupt the perceptual illusion of a talking head.

**Fine-grained Visual: Text Rendering Quality.** As indicated in Table 2, text rendering remains a significant bottleneck. Our analysis reveals a distinct performance dichotomy governed by text prominence and explicitness. Models generally succeed in rendering *explicitly prompted* text when the target string is short and occupies a dominant spatial region (e.g., a large movie title). However, performance degrades rapidly as text length increases or spatial resolution decreases, frequently resulting in "glyph collapse" or unintelligible gibberish. More critically, regarding *incidental text*—contextual writing not explicitly defined in the prompt (e.g., small print on a clapperboard)—we observe a **universal failure** mode across all evaluated models. Instead of generating coherent context-appropriate characters, models consistently hallucinate messy, graffiti-like scribbles.

**Fine-grained Visual: Facial Consistency.** Maintaining character identity across time remains a persistent challenge for all T2AV models. As shown in Table 2, even the top-performing model (Kling-V2.6) only achieves a consistency score of 57.33, while others hover around 48-54. We identify two primary degradation patterns: (1) **Temporal Identity Drift**: Identity features are highly unstable during discontinuities. When a character reappears after a shot transition, or undergoes large pose changes (e.g., turning their head), models often fail to recall the original face embeddings, effectively generating a new person. (2) **Crowd Degradation**: We observe a distinct "inverse scaling" law regarding

the number of faces. In multi-face scenarios (e.g., a cheering crowd), the rendering quality and stability of individual faces collapse significantly compared to single-portrait shots, resulting in distorted features and severe flickering.

**Fine-grained Audio: Pitch Accuracy.** A critical finding in our benchmark is that current T2AV models completely fail to understand musical notes. As shown in Table 2, all models achieve extremely low scores ($< 12/100$), indicating a lack of basic music theory knowledge. While models can correctly generate the *timbre* of an instrument, they cannot follow instructions regarding specific notes or pitch. When prompted to play a specific scale (e.g., "C Major") or chord sequence, models simply generate random notes that have no connection to the prompt.

**Fine-grained Audio: Speech Intelligibility & Coherence.** As reported in Table 2, Google's **Veo 3.1 series** demonstrates dominant performance in speech generation, with the *Quality* variant achieving a remarkable score of **96.09** and *Fast* at 94.53. This suggests that Veo has largely bridged the gap between video generation and TTS, maintaining high clarity even in complex scenes. However, significant limitations persist in other systems. We identify two primary failure modes: (1) **Hallucination in Contextual Speech**: When prompts imply speech without dictating a script (Incidental Mode), open-source models like Ovi (76.49) and LTX-2 frequently generate unintelligible gibberish or "alien languages." (2) **Partial Instruction Dropping**: In *Verbatim Mode*, even capable models often omit specific words or truncate sentences when long or complex dialogue is explicitly required.

**Physical Plausibility.** The evaluation results highlight significant deficits in how models model the physical world. First, in *Low-Level Kinematic Plausibility*, most models fail to surpass the passing threshold (a score of 4.0 in Video-Phy2). This indicates that the underlying physics of generated videos are often flawed, frequently exhibiting unnatural motion or object instability. Second, regarding *High-Level Causal Reasoning*, models demonstrate a lack of precise "world knowledge," leading to incorrect physical phenomena. For instance, in the prompt describing "sodium dropped into water," almost all models fail to correctly simulate the sodium *floating* on the water surface (due to density differences); instead, they often depict it sinking or simply changing color without the correct physical dynamics.

**Holistic Semantic Alignment.** Finally, when evaluating overall alignment, we observe that models frequently ignore specific visual and audio controls as the prompt becomes more complex. This issue is particularly severe in open-source models, which often fail to capture multiple constraints simultaneously. While proprietary models demonstrate a significant advantage (likely due to richer training data), they still struggle with *complex audio layering*. For instance, when a prompt requires multiple overlapping sounds—such as background music, footsteps, and speech occurring at the same time—even top-tier models tend to "drop" some audio elements, failing to generate a complete acoustic scene.

### 4.3. User Study

To validate our fine-grained evaluation framework, we recruited **10 expert raters** to annotate a shared subset of **85 tasks** across all six fine-grained dimensions. The subset was used to measure both human–metric correlation and inter-rater agreement; detailed protocols and interfaces are provided in Appendix C, with quantitative results in Tables 3 and 4.

Overall, our automated metrics show strong agreement with expert judgment on **five out of six** dimensions, with Pearson correlations of **0.9657** for *Text Rendering*, **0.8270** for *Facial Consistency*, **0.8300** for *Speech Intelligibility & Coherence*, **0.8290** for *Physical Plausibility*, and **0.8402** for *Holistic Semantic Alignment*. Inter-rater agreement is similarly high on these dimensions. The only weaker case is *Pitch Accuracy*, where both correlation (**0.5544**) and agreement ($\kappa = 0.3156$) are lower due to a floor effect: current T2AV systems perform extremely poorly on explicit pitch control, causing human ratings to cluster in a narrow low-score range.

### 4.4. Stability of the Evaluation

Beyond human alignment, we further assess the stability of our MLLM-assisted evaluation from two complementary perspectives: **run-to-run consistency** and **benchmark-scale robustness**.

First, we measure **run-to-run consistency** by repeating the full evaluation pipeline **3 times** on the same generated outputs, with detailed statistics reported in Table 5 in the appendix. The observed fluctuations are generally small across runs. For example, on Veo 3.1 Fast, the standard deviation is only **0.83** for Text, **0.08** for Music, **0.02** for Speech, and **0.28** for Holistic evaluation. LTX-2 shows similarly stable behavior across most dimensions. These results indicate that, although our framework includes an MLLM-based reasoning component, the resulting scores are stable in practice under repeated evaluation.

Second, we test whether the benchmark scale is sufficient for stable model comparison. Specifically, we repeatedly sample random prompt subsets at different ratios (**20%**, **40%**, **60%**, and **80%**) and recompute the overall normalized score. For each ratio, we repeat the sampling procedure **200 times**, with the full resampling curves provided in Figure 6 in the appendix. In both representative models, **Veo 3.1 Fast** and **LTX-2**, the subset-based estimates remain close to the full score, while the variance decreases steadily as the subset ratio increases. This indicates that AVGen-Bench provides stable model-level comparison under prompt subsampling, and that the full benchmark scale is sufficient for statistically meaningful evaluation.

Taken together, these results show that AVGen-Bench is not only human-aligned, but also stable with respect to repeated evaluation and prompt subsampling, supporting its use as a reliable benchmark for T2AV generation.

## 5. Conclusion

In this paper, we introduced **AVGen-Bench**, a task-driven framework for T2AV evaluation. Our results reveal a sharp dichotomy: while state-of-the-art models excel at **general audio-visual aesthetics**, creating cinematic content, they fail significantly at **fine-grained semantic control**. This is evidenced by low scores in tasks requiring precise pitch, text rendering, and physical logic. These findings suggest that current training paradigms based on coarse alignment are insufficient. Future research must prioritize finer-grained supervision to transition from probabilistic texture generators to physically grounded world models.

## Impact Statement

This work presents a benchmark and evaluation framework for text-to-audio-video generation. By identifying failures in fine-grained semantic control, speech coherence, physical plausibility, and audio-visual alignment, AVGen-Bench can help improve the reliability and controllability of generative

media systems.

The same progress may also increase risks associated with synthetic media, including misleading audiovisual content, impersonation, and unauthorized generation of realistic scenes or voices. Although this work does not introduce a new generation model, improved evaluation can indirectly accelerate model development. We encourage the use of AVGen-Bench alongside safeguards such as consent-aware data practices, provenance tracking, watermarking, and careful review in high-stakes applications.

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

## A. Additional Quantitative Validation Results

*Table 3.* **Human validation of fine-grained evaluation metrics.** Pearson correlation between automated scores and expert judgments on 85 shared tasks. Higher is better.

| Dimension | Protocol | Pearson ↑ |
|---|---|---|
| Text Rendering | Pointwise | 0.9657 |
| Pitch Accuracy | Pointwise | 0.5544 |
| Facial Consistency | Pairwise | 0.8270 |
| Speech Intelligibility & Coherence | Pairwise | 0.8300 |
| Physical Plausibility | Pairwise | 0.8290 |
| Holistic Semantic Alignment | Pairwise | 0.8402 |

*Table 4.* **Inter-rater agreement on the shared user-study subset.** Reliability across 10 expert raters. Pointwise dimensions use weighted Cohen's $\kappa$; pairwise dimensions use Cohen's $\kappa$. Higher is better.

| Dimension | Agreement Metric | Score ↑ |
|---|---|---|
| Text Rendering | Weighted Cohen's $\kappa$ | 0.9116 |
| Pitch Accuracy | Weighted Cohen's $\kappa$ | 0.3156 |
| Facial Consistency | Cohen's $\kappa$ | 0.8511 |
| Speech Intelligibility & Coherence | Cohen's $\kappa$ | 0.9272 |
| Physical Plausibility | Cohen's $\kappa$ | 0.8455 |
| Holistic Semantic Alignment | Cohen's $\kappa$ | 0.8909 |

| Model | Metric | Mean | Std. ↓ | Range |
|---|---|---|---|---|
| Veo 3.1 Fast | Text | 74.75 | 0.83 | 73.95–75.90 |
| | Face | 52.97 | 0.00 | 52.97–52.97 |
| | Music | 2.76 | 0.08 | 2.65–2.81 |
| | Speech | 94.48 | 0.02 | 94.46–94.51 |
| | Lo-Phy | 74.79 | 0.00 | 74.79–74.79 |
| | Hi-Phy | 70.73 | 1.70 | 69.16–73.09 |
| | Holistic | 85.47 | 0.28 | 85.06–85.68 |
| LTX-2 | Text | 26.91 | 0.59 | 26.29–27.71 |
| | Face | 45.54 | 0.00 | 45.54–45.54 |
| | Music | 7.57 | 1.24 | 5.88–8.82 |
| | Speech | 86.96 | 0.12 | 86.82–87.11 |
| | Lo-Phy | 81.45 | 0.00 | 81.45–81.45 |
| | Hi-Phy | 63.11 | 0.54 | 62.51–63.82 |
| | Holistic | 66.64 | 0.60 | 65.84–67.30 |

*Table 5.* **Repeated-run stability of the MLLM-assisted evaluation.** Mean, standard deviation, and value range over 3 repeated runs on the same generated outputs. Lower standard deviation indicates better stability.

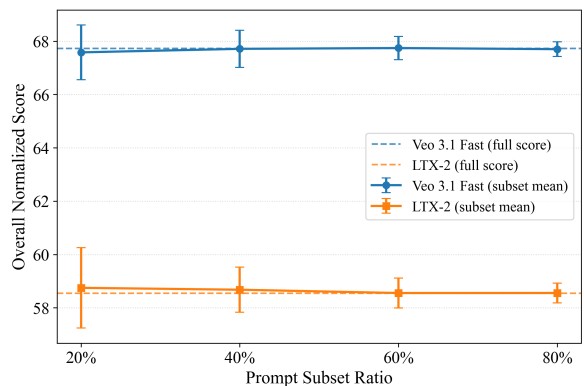

*Figure 6.* **Benchmark-scale robustness under prompt subset resampling.** Subset-based scores over 200 random resampling trials remain close to the full-benchmark scores, with variance decreasing as the subset ratio increases.

## B. Additional Qualitative Results

In this section, we provide extended qualitative examples to further illustrate the failure modes discussed in the main paper. We categorize these failures into three groups: (1) Text Rendering Failures (Figure 7), (2) Consistency and Speech Failures (Figure 8), and (3) Physical and Semantic Logic Failures (Figure 9).

## C. Human Evaluation Protocols and Interfaces

To ensure the reproducibility and rigorousness of our meta-evaluation, we developed a unified annotation platform using **Gradio**. We employed a **hybrid annotation strategy**, selecting the most appropriate protocol (Pairwise vs. Pointwise) based on the nature of the specific evaluation dimension.

The study covers all six fine-grained dimensions in AVGen-Bench: *Text Rendering*, *Pitch Accuracy*, *Facial Consistency*, *Speech Intelligibility & Coherence*, *Physical Plausibility*, and *Holistic Semantic Alignment*. We recruited **10 expert raters** and asked them to annotate a shared subset of **85 tasks**. This subset was used both for evaluating the correlation between our automated metrics and human judgments, and for measuring inter-rater agreement. For dimensions that require absolute judgment of a single output—namely *Text Rendering* and *Pitch Accuracy*—we used **pointwise scoring**. For dimensions that are more naturally assessed in relative terms—*Facial Consistency*, *Speech Intelligibility & Coherence*, *Physical Plausibility*, and *Holistic Semantic Alignment*—we used **pairwise comparison**. This hybrid design mirrors the structure of our automatic evaluation pipeline and allows us to assess both metric validity and annotation consistency under realistic conditions.

### C.1. Hybrid Annotation Strategy

**1. Pairwise Comparison for Subjective Quality (Speech & Semantic).** For dimensions where quality is often relative or nuanced—such as *Speech Quality* and *Holistic Semantic Alignment*—we utilized a **Blind A/B Testing** protocol (Figure 11a).

- **Rationale:** Determining "which voice sounds more natural" is cognitively easier and more consistent via side-by-side comparison than absolute scoring.

- **Mechanism:** Annotators are presented with two anonymized videos (randomized Left/Right order) and the strict prompt constraints. They must vote for the superior model or select "Tie". Notably, the interface explicitly displays *required speech lines* to force verification of verbatim adherence.

**2. Pointwise Scoring for Objective Correctness (Text Rendering).** Conversely, text rendering requires an absolute assessment of legibility and spelling correctness. A pairwise comparison might result in a "Tie" if both models produce gibberish, failing to capture the absolute failure. Therefore, we adopted a **Pointwise Protocol** (Figure 11b).

- **Rationale:** Text quality is objective (e.g., a typo is a typo). Absolute scoring allows us to quantify the exact success rate of each model.

- **Rubric:** We used a 3-point scale: **Good** (Fully legible and correct), **OK** (Minor artifacts but legible), and **Poor** (Illegible/Hallucinated/Missing).

**Prompted Text Rendering Failures**

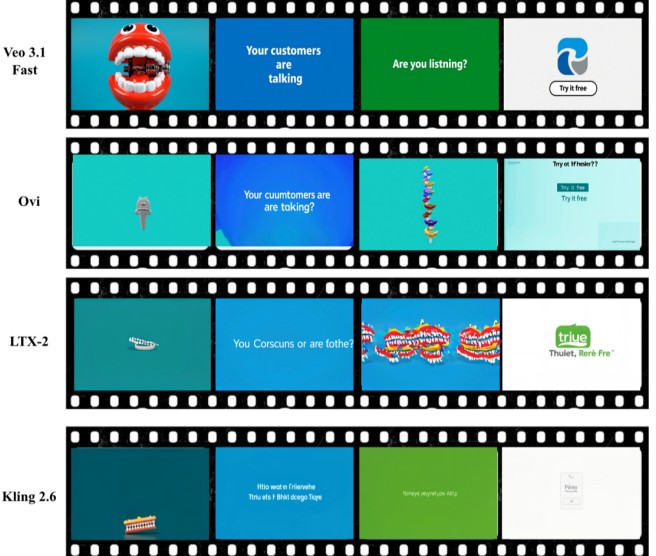

**Prompt**

A single wind-up chattering teeth toy clacks continuously against a solid teal background. The scene cuts to a blue screen displaying the white text **'Your customers are talking,'** abruptly followed by rows of multi-colored chattering teeth toys all moving at once, creating a loud chaotic mechanical clatter. A green screen appears with the text **'Are you listening?'** before cutting to a generic product logo and a **'Try it free'** button on a white background as the noise ceases.

**Prompt**

Four-shot high-tempo teaser with clean sync hits. Shot 1: Inside a bank vault, fluorescent hum and distant alarms; **a timer on a device beeps faster** as a thief whispers, "Eighty-seven seconds—move." Shot 2: Close-up of a glass cutter scoring a pane with a sharp scratch, then a suction cup pops as the circle lifts free, landing on a bass hit. Shot 3: Smash cut to a getaway car; engine revs, tires chirp, and the car fishtails out of a tight alley with gravel spraying and rattling off the chassis. Shot 4: A final slow-motion shot of a duffel bag hitting the pavement with a heavy thud as sirens surge; the title **EIGHTY-SEVEN SECONDS** slams onto black with a metallic logo sting.

**Incidental Text Rendering Failures**

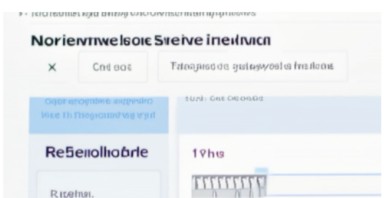
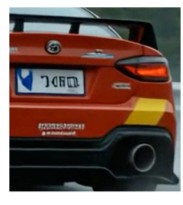
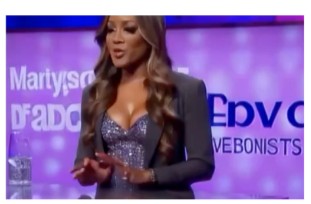
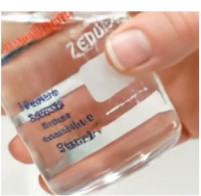

|  |  |  |  |
|---|---|---|---|
| **Website Content** | **Car license plate** | **Studio Backdrop** | **Beaker** |

*Figure 7.* **Extended Examples of Text Rendering Failures. Top (Prompted Text):** Models struggle with "glyph collapse" and layout errors when prompted with specific strings like "Your customers are talking" or "EIGHTY-SEVEN SECONDS". Even high-performing models like Veo 3.1 and Wan 2.6 often fail to render the text perfectly legible or place it on the correct object. **Bottom (Incidental Text):** A pervasive failure mode where models hallucinate gibberish for background text that was not explicitly prompted, such as website content, car license plates, or studio backdrops. This highlights a lack of "world knowledge" regarding how text naturally appears in real-world scenes.

## Face Inconsistency

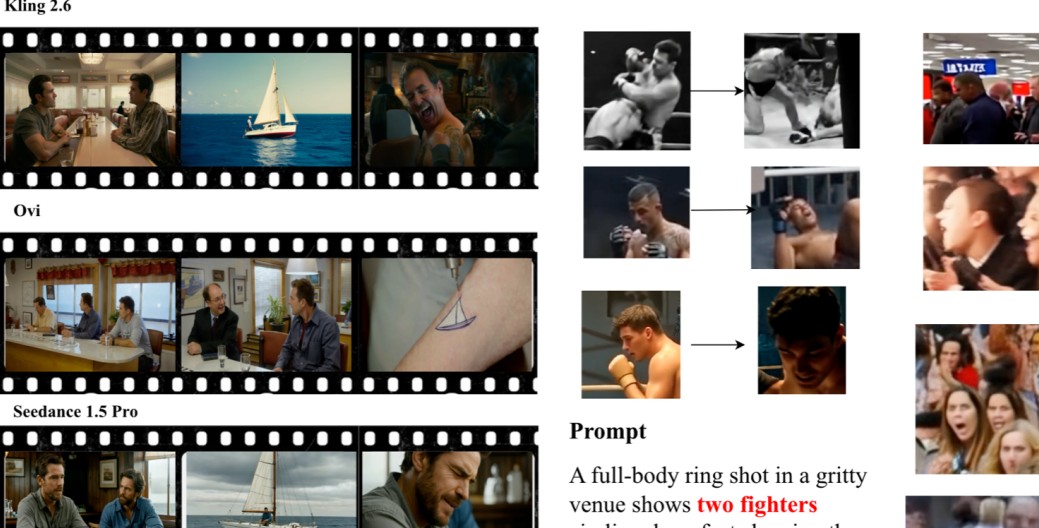

**Prompt**

**Two men** sit at a diner counter with soft ambient clatter, and one asks if the other is selling the boat. A clip shows **a man** sailing a boat as a voice asks, 'Are you selling my boat?' Back at the diner, **the man** asks how he can keep the memory of the boat. The video cuts to **him** screaming in a tattoo parlor as a buzzing tattoo machine etches a small sailboat onto his arm, the buzz rising and falling as the needle touches skin.

**Prompt**

A full-body ring shot in a gritty venue shows **two fighters** circling, bare feet slapping the canvas and the crowd rumbling. Punches land on gloves and forearms with dull thuds, then one fighter spins into a back elbow. The elbow connects with a sharp smack synced to the head snap, sweat spraying and pattering to the mat. The opponent staggers and drops, ropes creaking as the referee rushes in, and the audience noise surges.

**Face Rendering Failures When Multiple Faces are on-screen**

## Speech Generation Error

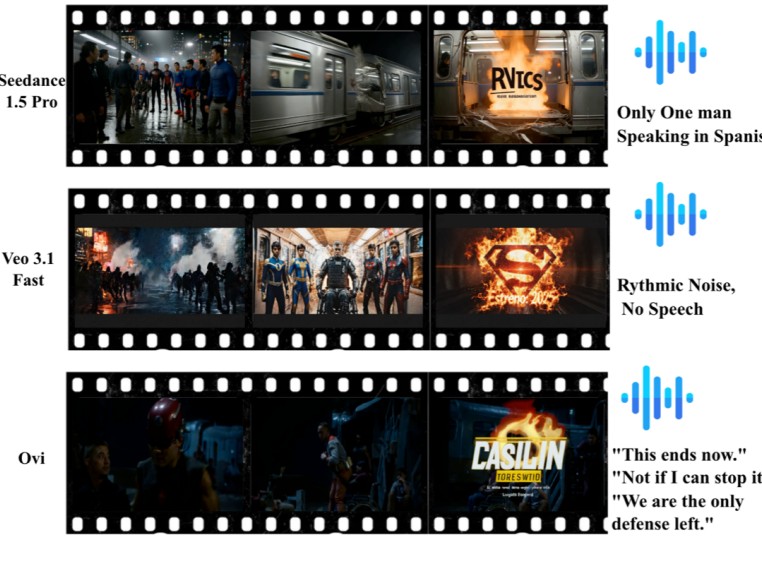

**Only One man Speaking in Spanish**

**Rythmic Noise, No Speech**

"This ends now."
"Not if I can stop it."
"We are the only defense left."

**Prompt**

A dark street scene shows a standoff between two groups with tense low rumble. A close-up of a man in a maroon helmet. A man in a wheelchair sits beside teenagers in superhero costumes. A subway train car is ripped apart by invisible forces with loud metal tearing and crashing, and soldiers tumble inside a moving structure amid heavy impacts. A fiery logo for a generic superhero film emerges from flames with a release date in Spanish, **while two male voices trade urgent lines and intense cinematic music drives the destruction.**

*Figure 8.* **Extended Examples of Face Inconsistency and Speech Generation Errors. Top (Face Inconsistency):** We observe two distinct patterns of identity loss: (1) *Identity Drift* across shot transitions, where a character's appearance changes significantly after a cut; and (2) *Crowd Degradation*, where faces in multi-person scenes (e.g., boxing audience) become distorted. **Bottom (Speech Generation):** Models frequently fail to adhere to linguistic or speaker constraints. Failures include generating the wrong language (e.g., Spanish instead of English), producing rhythmic noise instead of dialogue, or assigning dialogue to the wrong speaker count.

**Violation of Physical Laws**

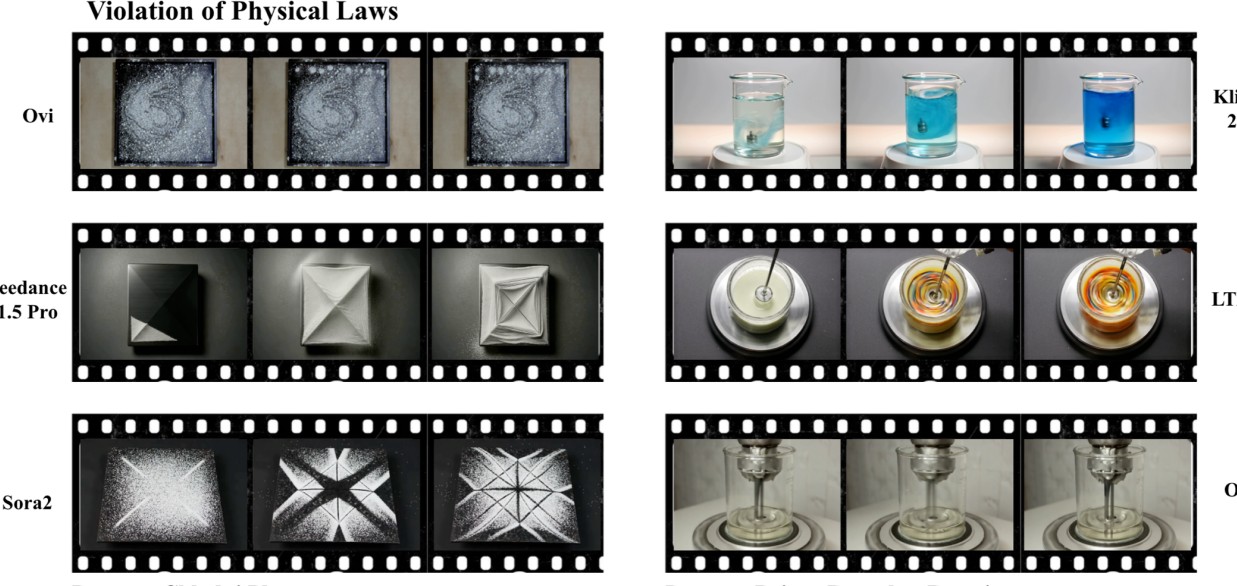

**Prompt: Chladni Plate**

A top-down view of a black square metal plate sprinkled evenly with fine white sand as a tone generator plays a pure sine wave that sweeps upward in pitch. As the plate begins to vibrate, the rising tone makes the sand suddenly jitter and chatter across the metal, then fall quiet as grains slide into crisp geometric nodal lines that sharpen and rearrange each time the pitch crosses a new resonance.

**Prompt: Briggs-Rauscher Reaction**

A high-speed time-lapse shows a beaker on a magnetic stirrer, the stir plate motor making a steady whir as a stir bar spins. The beaker contains a Briggs–Rauscher mixture (**hydrogen peroxide, potassium iodate, malonic acid, and a metal-ion catalyst with starch indicator**). While the vortex turns, the liquid repeatedly cycles through several distinct visible states in a rhythmic pattern, switching abruptly and then returning again and again as the stirring continues.

**Semantic Misalignment**

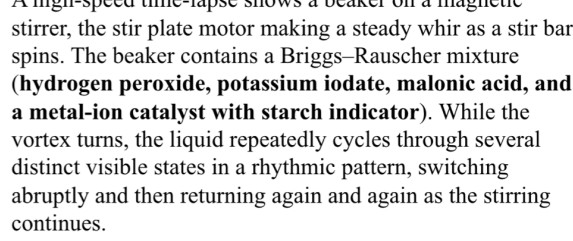

**Prompt: Vacation Rental Platform Ad**

A young boy **hits a beach ball** as a group of children runs past him and jumps into a swimming pool with loud splashes, while a voiceover states, 'We went on vacation with a toe dipper.' The camera follows the kids underwater as bubbles roar and feet kick past the lens, and the voiceover finishes, 'and left with a cannonballer.' Finally, the view **resurfaces** to show a laughing girl in the water as on-screen text reads 'Book your family home now.'

*Figure 9.* **Extended Examples of Physical Violations and Semantic Misalignment. Top (Violation of Physical Laws):** Models fail to simulate complex physical phenomena driven by sound. *Left (Chladni Plate):* Models fail to generate the correct geometric sand patterns corresponding to resonant frequencies. *Right (Chemical Reaction):* Models fail to depict the correct color oscillations or liquid dynamics in a Briggs-Rauscher reaction setup. **Bottom (Semantic Misalignment):** In complex multi-shot narratives (e.g., a vacation ad), models often miss key semantic constraints, such as specific actions ("hitting a beach ball") or correct text sequencing ("Book your family home now").

# Prompt

A top-down tutorial shot of a piano keyboard with a metronome. The player performs four block chords in sequence, one per bar: **C major, then G major, then A minor, then F major.** Each chord is pressed cleanly as a block (no arpeggio), held briefly, then released before the next chord.

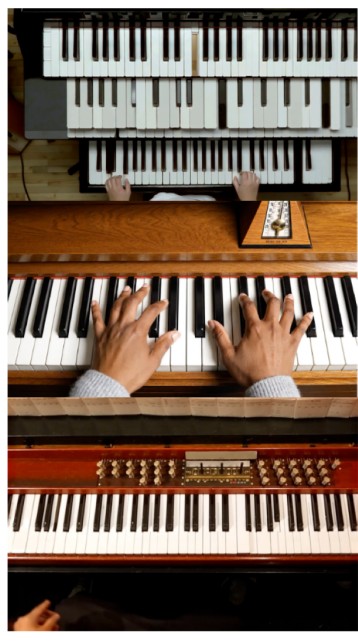

**Ovi:** The performance contains three of the four requested chords (C Major, A Minor, F Major) but in the wrong order and missing the G Major chord. The requirement for 'clean block chords' is also violated by significant melodic noise between the chord events.

**Veo 3.1 Fast:** The performance fails to provide block chords. The MIDI data contains only single notes or octaves played in a rapid, noisy sequence that does not resemble the requested C-G-Am-F progression in structure or timing.

**LTX-2:** The MIDI data fails to meet any of the prompt requirements. It contains a sequence of single notes (F#, G, F#, F, D#, D) rather than the requested sequence of four block chords (C, G, Am, F). No triads or major/minor chord qualities were detected.

# Prompt

A zoomed-in tutorial shot of a clean-tone electric guitar fretboard and picking hand. The player frets **a single note A4 and plucks it four times** with even timing, letting each note ring briefly. The pitch stays stable (no bend, no vibrato), and no other strings ring.

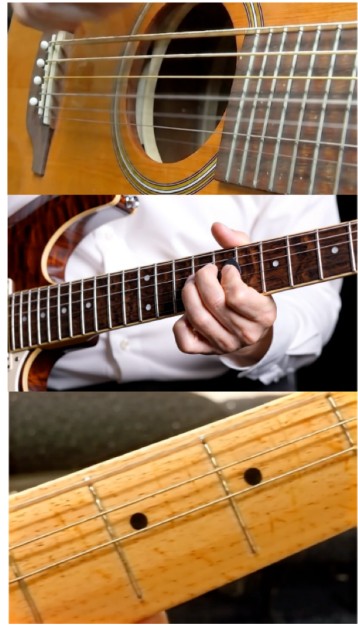

**Ovi:** The performance failed to provide a clean sequence of four A4 notes. Only two A4 notes were found, both of which were played simultaneously with other notes, and the recording contains a large number of additional notes not requested in the prompt.

**Veo 3.1 Fast:** The MIDI content completely fails to match the prompt. Instead of four isolated plucks of A4 with no other strings ringing, the data provides a complex sequence involving multiple pitches, chords, and several octaves, violating all core constraints.

**LTX-2:** The music content is a total mismatch. The prompt requested a single note (A4) plucked four times in isolation. The provided MIDI data contains a complex arrangement of chords and melodies using B, F#, D#, and other pitch classes, with no A4 notes present.

*Figure 10.* **Deep Dive into Pitch Accuracy Failures via Symbolic-Neural Verification.** We illustrate the disconnect between visual realism and acoustic logic in music generation. **Top (Piano):** The prompt strictly requests a "C-G-Am-F" chord progression. While models generate convincing visuals of hands on keys, the extracted MIDI data reveals that the audio contains wrong chords, random melodic noise, or chaotic note clusters, failing to follow basic music theory constraints. **Bottom (Guitar):** The prompt requests a specific single note (A4) plucked four times. Models fail to isolate the pitch, instead generating complex, unprompted chords or multi-string noise. This confirms that current T2AV models function as "texture generators" rather than grounded simulators of physical acoustic events.

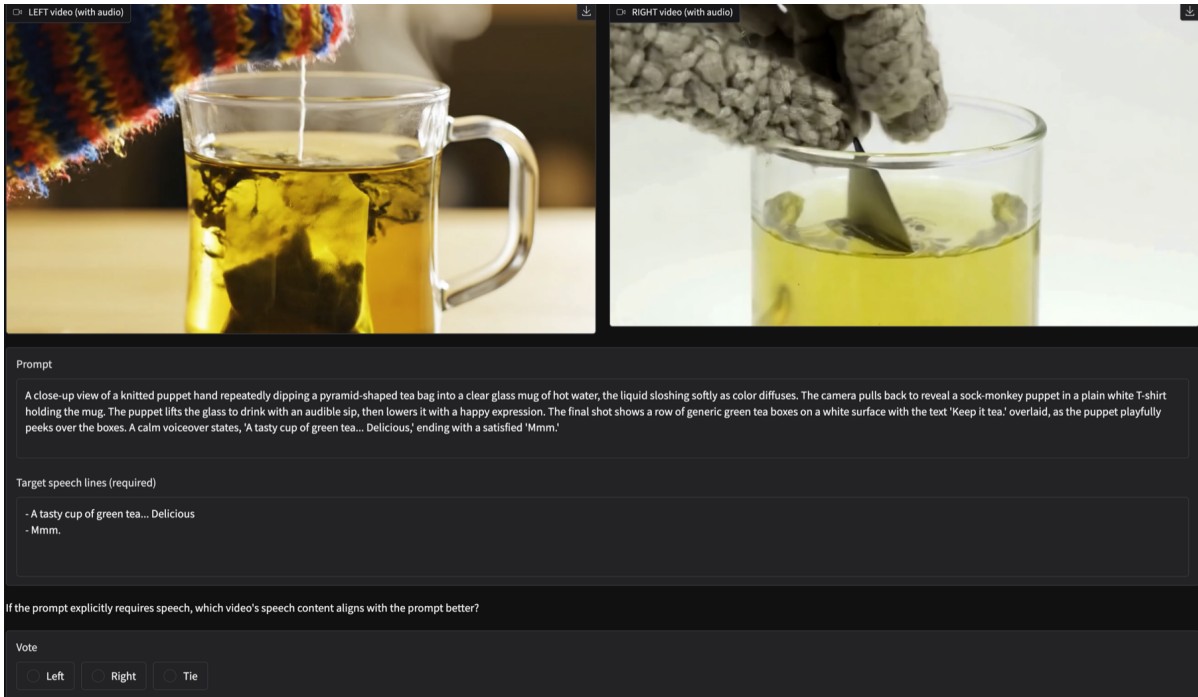

*(a)* **Pairwise Interface (Speech/Semantic):** Used for relative quality assessment. Features blind A/B testing with explicit constraint display.

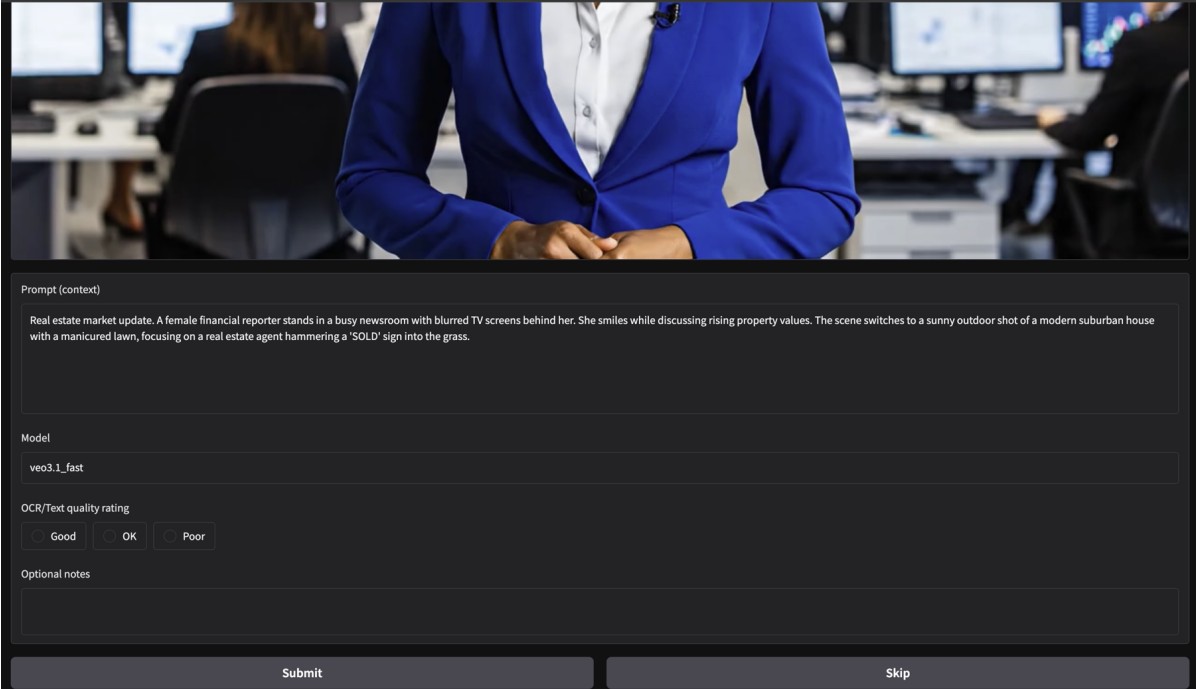

*(b)* **Pointwise Interface (Text Rendering):** Used for absolute quality assessment. Features a 3-point rubric (Good/OK/Poor) to judge objective legibility.

*Figure 11.* **Overview of the Custom Gradio Annotation Suite.** We tailored the annotation interface to the specific nature of the task. (a) For subjective dimensions, we enforce strict side-by-side comparison to reduce inter-rater variance. (b) For objective dimensions like text, we use absolute scoring to capture specific failure modes.

