# OpenReview forum: "AVGen-Bench: A Task-Driven Benchmark for Multi-Granular Evaluation of Text-to-Audio-Video Generation"
_ICML.cc/2026/Conference — ICML 2026 regular_

### Official Review · Reviewer_sQ73 · 2026-02-25

**Soundness:** 2
**Presentation:** 3
**Significance:** 3
**Originality:** 3
**Overall Recommendation:** 4
**Confidence:** 3

**Summary:**

This paper introduces AVGen-Bench, a task-driven benchmark for evaluating text-to-audio-video generation with diagnostic, fine-grained metrics beyond global perceptual scores. The benchmark covers diverse real-world prompt categories and evaluates models along multiple axes, including visual and audio quality, audio-visual synchronization, scene text rendering, facial identity consistency, pitch controllability, speech intelligibility/coherence, physical plausibility, and holistic semantic alignment. The evaluation pipeline combines specialist extractors with an MLLM-based reasoning/auditing component to verify whether prompt constraints are satisfied and to surface common failure modes.

**Compliance With Llm Reviewing Policy:**

Affirmed.

**Final Justification:**

My concerns have been largely addressed, and I am maintaining my positive score.

**Key Questions For Authors:**

1. In Verbatim mode with reference text available, objective metrics like WER are standard. Why choose an MLLM-based scorer here, and what is the specific advantage over the WER metric?

2. Are Scene Text Rendering, Facial Consistency, Pitch Accuracy, and Speech Intelligibility & Coherence evaluated for all prompts, or only for prompts that explicitly require those constraints? If conditional, what is the gating rule and coverage for each metric?

Other concerns are detailed in the weaknesses section.

**Limitations:**

yes

**Strengths And Weaknesses:**

### Strengths
1. Provides a task-driven benchmark that targets real user intents rather than only generic similarity metrics.
2. Uses multi-axis, failure-mode-oriented evaluation, helping diagnose *why* a system fails (text, face, speech, pitch, physics, etc.).
3. Combines specialist tools and structured verification, which is a practical way to scale evaluation for complex prompts.

### Weaknesses
1. The evaluation heavily depends on an MLLM for constraint verification and scoring, but the paper does not quantify judge accuracy, robustness , or systematic bias.
2. The exact MLLM prompts, scoring guidelines, and decision rules are not clearly provided, making the fine-grained results hard to reproduce or audit. For example, “verbatim vs contextual” modes in Speech Intelligibility & Coherence and their concrete scoring standards are not explicitly specified.
3. Using only two participants limits statistical power and increases susceptibility to individual bias; it is unclear whether reported trends would hold with a broader rater pool and inter-rater reliability analysis.
4. Modules relying on ASR/AMT can degrade under background noise; these errors may cascade into the MLLM’s final judgment, but the paper does not provide an uncertainty analysis.

---

> ### Author Rebuttal · Authors · 2026-03-31
>
> We thank the reviewer for the careful reading. Below we address the concerns.
>
> ### W1: Reliability of MLLM-based evaluation
> We agree that MLLM-based evaluation should be validated carefully; we therefore assess **human alignment**, **inter-rater stability**, and **repeated-run robustness**. Beyond the 3 dimensions in **Section 4.3**, we additionally conducted a **6-dimension, 10-rater** human study.
>
> | Dimension | Pearson |
> |---|---:|
> | Text | 0.9657 |
> | Pitch | 0.5544 |
> | Facial Consistency | 0.8270 |
> | Speech | 0.8300 |
> | Physical Plausibility | 0.8290 |
> | Holistic Semantic | 0.8402 |
>
> This shows strong human-model alignment on **5/6 dimensions**. The weaker case, **Pitch Accuracy**, mainly reflects a **floor effect**: current T2AV systems perform extremely poorly on this dimension (best model only **11.25/100**), and human ratings are almost entirely concentrated at **1–2**.
>
> Inter-rater agreement on a fixed **85-task, 10-rater** subset is also strong: **Text Rendering** (weighted κ **0.9116**), **Pitch Accuracy** (weighted κ **0.3156**), **Facial Consistency** (Cohen’s κ **0.8511**), **Speech** (Cohen’s κ **0.9272**), **Physical Plausibility** (Cohen’s κ **0.8455**), and **Holistic** (Cohen’s κ **0.8909**).
>
> Repeated-run fluctuation is also small. For **Veo 3.1 Fast**, std values are **0.8347 / 0.0000 / 0.0780 / 0.0202 / 1.6987 / 0.2844** for **Text / Face / Pitch / Speech / Hi-Phy / Holistic**; **LTX-2** shows similarly small fluctuations. Overall, the MLLM-assisted evaluation is well aligned with human judgment, stable across raters, and robust across repeated runs.
>
> ### W2: Reproducibility of prompts and scoring rules
> We agree that the main issue here is **reproducibility and auditability**. Our fine-grained evaluation is not unconstrained free-form MLLM judging; e.g., **Scene Text Rendering** already follows an explicit **detect–aggregate–verify** pipeline in the paper. We also agree that the current draft does not make the **MLLM prompt templates**, **scoring logic**, and the distinction between **Verbatim** and **Contextual** modes in **Speech Intelligibility & Coherence** sufficiently explicit. Concretely, **Verbatim** checks faithful delivery when a reference utterance exists, while **Contextual** checks intelligibility/coherence when no exact reference exists. We will release the **full evaluation codebase**, including the **MLLM prompt templates**, **decision rules**, and the corresponding **prompt set / generated videos**.
>
> ### W3: Broader human validation
> As detailed in **W1**, we additionally conducted a **6-dimension, 10-rater** human study together with **inter-rater agreement** analysis. These results show strong human-model alignment on **5/6** dimensions and high inter-rater stability on most dimensions, so the validation is not limited to two participants.
>
> ### W4: Error propagation from ASR/AMT
> We agree that uncertainty propagation from upstream modules is relevant in any hybrid evaluation pipeline. The ASR and AMT components we use are established specialist models whose robustness has already been studied in their original papers, and our prompt design does **not** intentionally place target speech/music under extreme background noise. If a T2AV model itself generates excessive noise that corrupts speech or music, this should be counted as part of the model’s generation failure rather than merely an evaluation artifact.
>
> ### Q1: Why not use WER in Verbatim mode?
> We agree that **WER** is standard for pure transcription matching, but our setting is more complex: **Verbatim** and **Contextual** requirements may coexist in the same video. WER would then require isolating which part of the generated speech corresponds to the verbatim target, which is brittle in realistic T2AV outputs. By contrast, an MLLM-based scorer can jointly consider the **reference text**, **generated speech/transcript**, and **video context**, and thus better handle mixed verbatim/contextual settings.
>
> ### Q2: Gating rules and metric coverage
> These metrics are **conditional rather than uniformly applied to all prompts**:
> - **Text Rendering:** OCR extracts candidate text; the MLLM classifies it as **required / incidental / noise / none**. Official scoring averages only over **required + incidental** cases.
> - **Speech Intelligibility & Coherence:** ASR first produces a transcript; the MLLM then evaluates it jointly with the prompt and video. This metric is scored for **all videos**, since unexpected speech should also be penalized.
> - **Facial Consistency:** evaluated only for prompts with **human faces / identity-consistent appearance**.
> - **Pitch Accuracy:** evaluated only for prompts with **explicit pitch-control requirements**.
>
> For **Veo 3.1 Fast** as a representative example, the scored coverage is: **Text Rendering 86**, **Speech 234**, **Facial Consistency 103**, and **Pitch Accuracy 17**, with minor deviations due to a small number of closed-source generations blocked by safety filtering.

---

> > ### Author Rebuttal · Reviewer_sQ73 · 2026-04-03
> >
> > Thank you to the authors for their responses in the rebuttal. I hope that the relevant clarifications provided in the rebuttal can be incorporated into the revised version.

---

> > > ### Author Response · Authors · 2026-04-04
> > >
> > > We thank the reviewer again for the positive follow-up and for recognizing that the main concerns have been fully addressed.
> > >
> > > We are very grateful for the reviewer’s positive assessment and thoughtful feedback throughout the discussion. In the revised version, we will incorporate all of the clarifications raised in the rebuttal into the paper, including the details on human validation, reproducibility, scoring logic, and metric coverage.
> > >
> > > To further support the community, we also commit to open-sourcing the key benchmark resources, including the dataset-construction code, the evaluation and validation code, the benchmark prompt/test set, and the complete evaluation results for all tested models.
> > >
> > > We sincerely appreciate the reviewer’s recognition of the work, and we hope AVGen-Bench can serve as a useful resource for the community and help promote continued progress in T2AV evaluation.
> > >
> > > Thank you again for your thoughtful evaluation and valuable suggestions.

---

### Official Review · Reviewer_7cEX · 2026-03-12

**Soundness:** 3
**Presentation:** 4
**Significance:** 2
**Originality:** 2
**Overall Recommendation:** 4
**Confidence:** 3

**Summary:**

This paper introduces AVGen-Bench, a task-driven benchmark for evaluating text-to-audio-video generation. The benchmark aims to address limitations in existing evaluations that treat audio and video independently or rely mainly on embedding similarity metrics. The authors propose a multi-granular evaluation framework combining lightweight specialist models and MLLMs to assess several aspects of generation quality, including perceptual quality, semantic correctness, and cross-modal alignment.

**Compliance With Llm Reviewing Policy:**

Affirmed.

**Final Justification:**

The authors’ rebuttal has largely addressed my concerns, and I consider the paper to be borderline acceptable.

**Key Questions For Authors:**

1. Have the authors validated the MLLM-based evaluation against human judgments? If so, what is the correlation or agreement level?
2. How large is the benchmark in terms of prompts and generated samples? Is the scale sufficient to produce statistically meaningful comparisons between models?
3. Will the authors release the full prompt set, evaluation scripts, and model configurations to enable reproducibility?

**Limitations:**

yes

**Strengths And Weaknesses:**

Strengths:
1. Evaluation of T2AV generation is indeed an emerging and important problem as multimodal generative models become increasingly capable. A unified benchmark targeting both audio and video is valuable for the community.
2. The benchmark attempts to design prompts covering multiple realistic scenarios, which is helpful for probing the capabilities of generative systems in more practical contexts.
3. The paper provides useful insights into the failure modes of current T2AV systems, such as weaknesses in speech coherence, text rendering, and musical pitch control.
Weaknesses:
1. The core contribution mainly lies in dataset construction and evaluation design. The proposed evaluation framework largely combines existing components (specialist metrics + MLLM-based judging), and the novelty compared with recent multimodal evaluation benchmarks appears somewhat limited.
2. The paper does not provide enough details about how the prompts and evaluation tasks are curated. For example: how prompts are selected or generated; how diversity and difficulty are controlled. These factors are important for assessing the reliability of the benchmark.
3. The paper should more clearly differentiate itself from recent audio-video evaluation benchmarks and discuss advantages over them in terms of coverage, metric reliability, and evaluation granularity.

---

> ### Author Rebuttal · Authors · 2026-03-31
>
> ### W1.  Novelty.
> While our framework adopts a hybrid design (specialist models + MLLM), our key contribution lies in a task-driven and T2AV-specific evaluation paradigm, rather than new standalone components.
>
> (1) T2AV-tailored fine-grained metrics.
> We introduce capability-oriented dimensions such as Text Rendering, Facial Consistency, Pitch Accuracy, Speech Intelligibility, Physical Plausibility, and Holistic Alignment, enabling more comprehensive and diagnostic evaluation of cross-modal semantic control .
>
> (2) Task-driven prompt design.
> Prompts are derived from 11 real-world scenarios and user intents, decoupled from evaluation metrics, making them more realistic than prior metric-driven constructions .
>
> (3) Tailored hybrid pipeline.
> We combine small-model parsing + MLLM reasoning, balancing accuracy and efficiency, and avoiding the coarse granularity and high cost of MLLM-only evaluation.
>
> In summary, our novelty lies in advancing T2AV evaluation from coarse, metric-driven scoring to task-driven, fine-grained diagnosis, with improved coverage, realism, and practicality.
>
> ---
>
> ### W2. More details.
>
> We agree that the prompt/task curation process should be made more explicit. Our benchmark contains **235 prompts across 11 subcategories**; prompt themes were jointly determined by **8 experts**, and **4 experts** were involved in filtering candidate prompts. The filtering criteria are **Complexity, Clarity,** and **Diversity**.
>
> We adopt category-specific prompt strategies instead of a single template, including underspecified prompts for physics/chemistry, multi-shot scripts for movie trailers, and Gemini-based reverse captioning of YouTube ads. These designs improve diversity, difficulty control, and reliability.
>
> ---
>
> ### W3. Differentiation.
> We agree this distinction should be clearer. **Section 2.2** already contrasts AVGen-Bench with prior benchmarks, many of which either isolate audio/video, focus on coarse quality, or rely on embedding-based matching that cannot verify fine-grained constraints. Regarding additional recent benchmarks: **TAVGBench** mainly relies on coarse distribution/embedding-based metrics; **MTAVG-Bench** focuses on **multi-talker dialogue-centric** AV generation; and **VABench/T2AV-Compass** are closer to our setting but mainly follow a **small-model / large-model separation**. In contrast, AVGen-Bench is distinguished by **coverage** (11 real-world categories and fine-grained controllability dimensions), **metric reliability** (specialist parsing + structured MLLM reasoning), and **evaluation granularity** (diagnosing *why* a model fails, not only whether overall quality is high).
>
> ---
>
> ### Q1 Human Alignment
> Yes. **Section 4.3** already includes a small-scale user study on 3 dimensions; we additionally conducted a **larger-scale human study over all 6 fine-grained dimensions with 10 expert raters**.
>
> | Dimension | Pearson |
> |---|---:|
> | Text Rendering | 0.9657 |
> | Pitch Accuracy | 0.5544 |
> | Facial Consistency | 0.8270 |
> | Speech Intelligibility & Coherence | 0.8300 |
> | Physical Plausibility | 0.8290 |
> | Holistic Semantic Alignment | 0.8402 |
>
> This shows **strong human-model alignment on 5/6 dimensions**. The weaker case, **Pitch Accuracy**, reflects a **floor effect**: current T2AV systems perform extremely poorly on this dimension (best model only **11.25/100**), and human ratings are almost entirely concentrated at **1–2**.
>
> We also conducted an inter-rater agreement study on a fixed shared subset of 85 tasks with all 10 raters: Text Rendering (weighted κ 0.9116, α 0.9252), Pitch Accuracy (weighted κ 0.3156, α 0.3650), Facial Consistency (agreement 0.9141, Cohen’s κ 0.8511, α 0.8523), Speech Intelligibility & Coherence (agreement 0.9524, Cohen’s κ 0.9272, α 0.9274), Physical Plausibility (agreement 0.8998, Cohen’s κ 0.8455, α 0.8511), and Holistic Semantic Alignment (agreement 0.9383, Cohen’s κ 0.8909, α 0.8930). Overall, the MLLM-assisted evaluation is well aligned with expert judgment and the human ratings themselves are highly stable on most dimensions.
>
> ---
>
> ### Q2 Scale
> Our benchmark contains **235 prompts** and evaluates **9 T2AV models/pipelines**, with **one audio-video sample per prompt per model**, yielding **up to 2,115 generated samples** (a small number are skipped when closed-source systems refuse generation due to built-in safety filtering).
>
> To test scale sufficiency, we repeatedly sampled **80/60/40/20%** prompt subsets (**200 times each**) and recomputed an overall normalized score. For **Veo 3.1 Fast**, the std is **0.2779 / 0.4312 / 0.7000 / 1.0251**; for **LTX-2**, **0.3738 / 0.5620 / 0.8445 / 1.5102**. Thus, score fluctuation remains small under random subsampling, indicating that the benchmark scale is sufficient for statistically meaningful comparison.
>
> ---
>
> ### Q3 Reproducibility
> Yes. We plan to release the **full prompt set**, the **complete evaluation codebase**, the **generated videos**, and the corresponding **model/pipeline configurations**.

---

> > ### Author Rebuttal · Reviewer_7cEX · 2026-04-04
> >
> > Thank you for your response. It has addressed my concerns, and I will consider revising my score accordingly.

---

> > > ### Author Response · Authors · 2026-04-04
> > >
> > > We thank the reviewer again for the positive follow-up and thoughtful consideration of our rebuttal.
> > >
> > > We are especially encouraged that the main concerns have been addressed. In the final revision, we will make these points more explicit in the paper, including the benchmark curation process, the expanded human-validation results, the scale analysis, and the reproducibility details.
> > >
> > > To further support the community, we commit to **open-source the full benchmark resources**, including:
> > > - the code used for dataset construction,
> > > - the full evaluation and validation code,
> > > - the benchmark prompt/test set,
> > > - and the complete evaluation results for all tested models.
> > >
> > > Beyond the initial release, we commit to **actively maintain the benchmark** by continuously evaluating newly emerging T2AV models, so that AVGen-Bench can remain useful as the field evolves.
> > >
> > > More broadly, our goal is to provide a benchmark that is not only reproducible, but also practically useful for both the research community and real-world T2AV development. We hope these efforts help clarify that AVGen-Bench is intended as a **timely contribution** to a rapidly advancing area.
> > >
> > > We sincerely thank the reviewer for the time and careful suggestions throughout the discussion. We would be very grateful if, in light of this timely contribution, the reviewer would consider a **positive final assessment** as a recognition of the value of our work.
> > >
> > > Thank you again for your constructive feedback and thoughtful evaluation.

---

### Official Review · Reviewer_yQC4 · 2026-03-12

**Soundness:** 3
**Presentation:** 3
**Significance:** 3
**Originality:** 3
**Overall Recommendation:** 4
**Confidence:** 3

**Summary:**

This paper introduces AVGen-Bench, a task-driven benchmark designed to address the gap in evaluating fine-grained behaviors of Text-to-Audio-Video (T2AV) generation. The benchmark contains prompts across 11 real-world categories,  combining lightweight specialist models (PaddleOCR, SyncNet, Basic-Pitch) to construct the evaluation suite. The authors then reveal a gap between strong
audio-visual aesthetics and weak fine-grained semantic control, highlighting critical challenges in text, speech, and
physical reasoning

**Compliance With Llm Reviewing Policy:**

Affirmed.

**Final Justification:**

I will keep my positive score

**Key Questions For Authors:**

no

**Limitations:**

yes

**Strengths And Weaknesses:**

**Strength**

This paper tackles an important problem in benchmarking Text to Audio-Video Generation.  Current benchmarks for Text to Audio-Video generation are lacking in jointly evaluating the audio and visual quality of generated outputs especially when user specified visual and acoustic requirements. The authors demonstrated that their benchmark has high correlation with human judgement across three evaluation dimension.

Overall, the writing is clear and easy to follow.

**Weakness**

 While the authors argue that decoupling prompt curation from evaluation is a strength, it may lead to unsolvable evaluation tasks. For example, prompts derived from "genuine user needs" may contain requirements that the current specialist models simply lack the resolution to detect. Could the author add some clarification on this?

The task prompts are curated from GPT-5.2 and filtered for complexity, clarity, and diversity. Could the authors provide more detail on exactly how the prompts are filtered? How many person are involved in the prompt filtering process? Could there be a possibility of human subjective bias?

---

> ### Author Rebuttal · Authors · 2026-03-31
>
> We thank the reviewer for the positive assessment of the paper, especially for recognizing the importance of jointly evaluating audio-visual generation and the clarity of the writing. We also appreciate the reviewer’s constructive questions regarding the decoupling of prompt curation from evaluation and the prompt filtering process. Below we clarify these points.
> ### W1 Why decouple prompt curation from evaluation design?
> We thank the reviewer for raising this important point. Our motivation for decoupling prompt curation from evaluation design is that prompts should remain as close as possible to real user needs, so that the benchmark reflects the actual challenges encountered in realistic T2AV generation scenarios.
>
> In practice, our evaluation already covers many dimensions where current models empirically fail, such as Text Rendering, Speech Intelligibility & Coherence, Pitch Accuracy, Facial Consistency, and Physical Plausibility. At the same time, we acknowledge that no benchmark can exhaustively cover every possible failure mode. For example, some issues such as anomalous human generation (e.g., six fingers) may not yet be explicitly scored as a standalone dimension, but they can still be revealed through qualitative analysis or case studies.
>
> We believe this is still preferable to an evaluation-derived prompt design, where prompts are restricted to only what existing metrics can already measure cleanly. Such a design risks hiding important real-world failure modes altogether, and would provide less insight for future model development. In contrast, a task-driven prompt design better exposes the gap between realistic user demands and current model capabilities, which is precisely the goal of AVGen-Bench.
>
>
> ### W2 Prompt curation, filtering, and bias control
> We thank the reviewer for this helpful question. We agree that the prompt curation pipeline should be described more explicitly.
>
> Our benchmark contains **235 prompts** across **11 subcategories**, with the following distribution:
>
> | Subcategory | # Prompts |
> |---|---:|
> | News | 20 |
> | Ads | 20 |
> | Movie Trailer | 20 |
> | ASMR | 20 |
> | Musical Instrument Tutorial | 35 |
> | Cooking | 20 |
> | Gameplays | 20 |
> | Physical Experiment | 20 |
> | Chemical Reaction | 20 |
> | Animals | 20 |
> | Sports | 20 |
>
> Regarding the curation process, the **prompt themes/categories were jointly determined by 8 experts**, and after GPT-generated candidate prompts were produced for each category, **4 experts** were involved in the filtering stage.
>
> The filtering criteria are:
> - **Complexity**: whether the prompt contains meaningful multi-constraint or multi-step generation requirements;
> - **Clarity**: whether the prompt is specific enough to support reliable evaluation;
> - **Diversity**: whether the retained prompts cover varied scenarios rather than near-duplicate cases.
>
> We also took several steps to reduce subjective bias in prompt construction. Importantly, we do **not** use a single uniform prompt-generation template across all categories. Instead, the prompting strategy is adapted to the characteristics of each task domain. For example:
>
> - For **Physical Experiments** and **Chemical Reactions**, we use an **underspecified prompting strategy**, where the expected physical outcome is intentionally not stated explicitly, so that the benchmark tests whether the model can generate physically consistent results rather than simply copy a stated answer.
> - For **Movie Trailers**, we ask GPT to generate **multi-shot prompts** that mimic realistic trailer-style scene transitions and composition.
> - For **Commercial Ads**, we do not rely on GPT prompt writing; instead, we use **Gemini 3 Pro** to reverse-caption popular YouTube **bumper ads**, followed by anonymization.
>
> These category-specific design choices, detailedly described in **Section 3.1**, help ensure that prompt sources and structures are sufficiently diverse, and reduce the risk of bias from using a single prompt template or a single generation style.
>
>
> Overall, AVGen-Bench is designed to remain task-driven while still carefully curated, so that the benchmark captures realistic user-facing challenges without collapsing into either arbitrary prompt collection or metric-driven prompt engineering.

---

> > ### Author Rebuttal · Reviewer_yQC4 · 2026-04-04
> >
> > Thanks for the rebuttal. I will keep my positive score.

---

> > > ### Author Response · Authors · 2026-04-06
> > >
> > > Thank you again for the follow-up. We are very encouraged that our rebuttal has fully resolved the concerns, and we sincerely appreciate the reviewer’s positive assessment and continued support.
> > >
> > > In the final revision, we will incorporate the clarifications discussed in the rebuttal and will open-source the benchmark resources to improve reproducibility and benefit the community.

---

### Official Review · Reviewer_Go7R · 2026-03-18

**Soundness:** 3
**Presentation:** 2
**Significance:** 3
**Originality:** 3
**Overall Recommendation:** 4
**Confidence:** 4

**Summary:**

This paper introduces AVGen-Bench, a benchmark for text-to-audio-video (T2AV) generation. The main claimed contributions are: a task-driven prompt set spanning 11 real-world categories, and a hybrid multi-granular evaluation framework that combines specialist tools with MLLMs to score uni-modal quality, cross-modal synchronization, and fine-grained semantic controllability such as text rendering, facial consistency, musical pitch accuracy, speech intelligibility, physical plausibility, and holistic prompt alignment. The experiments evaluate several commercial and open-source T2AV systems and conclude that current models have strong aesthetics but weak fine-grained semantic control.

**Compliance With Llm Reviewing Policy:**

Affirmed.

**Key Questions For Authors:**

1. Why not use the same model for the Professional Media Production (Commercial Ads and Movie Trailers)?
2. Missing reference for Figure 3, and Vis should be explicitly mentioned in the section 4.2 for better understanding.
3. The manuscript’s comparison set appears selective. How does AVGen-Bench differ concretely from recent benchmarks such as TAVGBench, VABench, T2AV-Compass, and MTAVG-Bench, beyond having longer prompts or a different combination of metrics?
4. Many of the proposed metrics depend on Gemini for extraction or final scoring. How stable are these scores across prompt templates, repeated runs, or judge substitutions?
5. Why is the user study limited to only three dimensions and two expert raters?
6. Please make the benchmark size, category balance, and sample distribution explicit and prominent in the main paper.

**Limitations:**

The novelty is not yet convincingly established against the broader T2AV benchmark landscape, where several recent works already provide multi-dimensional, task-aware, or MLLM-assisted evaluation; as a result, the present contribution feels closer to a targeted extension than to a clearly new benchmark paradigm. The human validation is also too limited relative to the scope of the claims, since only three dimensions are studied and only two expert raters are used.

**Strengths And Weaknesses:**

Strengths
1. Evaluation for T2AV generation is genuinely underdeveloped relative to model progress, and the paper correctly argues that audio-visual generation should not be assessed only through coarse aesthetic or embedding-based similarity. The benchmark’s focus on joint correctness rather than separate audio and video quality is well motivated.
2. The paper does not stop at general quality or synchronization, but explicitly probes text rendering, speech coherence, pitch accuracy, facial consistency, physical plausibility, and holistic semantic alignment. That is a more actionable failure analysis than many earlier benchmark papers provide, and several of these axes are practically relevant to real user prompts.
3. Organizing prompts into professional media, creator economy, and world simulation is a better framing than a purely metric-driven collection. The prompt design choices for music constraints and underspecified physical reasoning are especially interesting because they try to test capabilities that models often fake aesthetically while failing semantically.
4. The results show a clear pattern: current systems score well on visual and audio aesthetics, but much worse on fine-grained control, with especially poor results for music pitch accuracy and only modest facial consistency or physical plausibility. Even if the benchmark itself is not yet fully mature, these observations are valuable to the community.

Weaknesses
1. The paper’s novelty claim is currently stronger than the evidence supports. The manuscript compares mainly against VBench, TTA-Bench, JavisBench, Harmony-Bench, VerseBench, and UniAVGen in Table 1, and emphasizes higher prompt complexity and more metrics. What about other T2AV benchmarks, e.g.,TAVGBench, VABench, T2AV-Compass, and MTAVG-Bench?
2. Many of the most important metrics rely on Gemini as a reasoning component: extracting music constraints, inferring expected physics outcomes, verifying speech coherence, determining character counts, and scoring holistic alignment. This may be practical, but it also introduces substantial evaluator subjectivity and instability. The paper does not thoroughly study prompt sensitivity, model-judge variance, or agreement between different MLLM evaluators.
3. The user study covers only three dimensions and uses only two expert raters. That is not enough to convincingly validate a benchmark that makes broad claims about ten dimensions and relies extensively on complex automated / MLLM-based scoring pipelines.
4. The main text snippets do not make the benchmark size especially prominent or easy to verify.
5. Missing references to some figures and inconsistencies in the manuscript of abbreviation usages.
6. Poor figure qualities with some overlapping, alignments and missing prompt usages.

---

> ### Author Rebuttal · Authors · 2026-03-31
>
> We thank the reviewer for the careful reading and constructive feedback. Below we address the main concerns briefly.
>
> ### W1 \& Q3 Novelty and distinction from recent T2AV benchmarks
> We agree that our distinction from recent T2AV benchmarks should be made more explicit. Our novelty lies not in longer prompts or more metrics, but in a different **benchmark formulation** based on **task-driven prompts** and **fine-grained joint audio-video semantic evaluation**.
>
> Compared with recent benchmarks, **TAVGBench** mainly relies on coarse distribution/embedding-based metrics, **MTAVG-Bench** focuses on **multi-talker dialogue-centric** AV generation, and **VABench/T2AV-Compass** mainly follow a **small-model / large-model separation**. In contrast, AVGen-Bench uses a tighter **small-model parsing + large-model reasoning** strategy; e.g., for **Text Rendering**, we first use OCR to extract text and then use Gemini to verify rendering accuracy and contextual appropriateness.
>
> More broadly, AVGen-Bench is distinguished by **task-driven prompt formulation**, **fine-grained controllability diagnosis**, and **joint audio-video semantic evaluation** rather than only overall quality or embedding similarity. This also addresses **Q3**: the difference is not merely longer prompts or more metrics, but a different benchmark formulation for joint audio-video semantic evaluation.
>
> ### W2 \& Q4 Stability of Gemini-based evaluation
> We agree that MLLM-based evaluation should be validated carefully.
>
> - **Why Gemini:** our benchmark requires **joint audio-video evaluation in one unified pipeline**, and the **Gemini series is currently the only mainstream MLLM that reliably supports synchronized audio-video reasoning**.
>
> - **Stability:** We also ran a **3-run repeated-evaluation test** on **Veo 3.1 Fast**. The resulting fluctuations are small: the standard deviations are **Text 0.8347**, **Facial Consistency 0.0000**, **Pitch Accuracy 0.0780**, **Speech Intelligibility & Coherence 0.0202**, **Hi-Phy 1.6987**, and **Holistic Semantic Alignment 0.2844**. This indicates that the Gemini-based evaluation pipeline is **stable in practice**.
>
> ### W3 \& Q5 Broader human validation and inter-rater reliability
> Section 4.3 already reports human-model correlation on 3 representative dimensions. We additionally conducted a **larger-scale human study covering all 6 fine-grained dimensions with 10 expert raters**.
>
> | Dimension | Pearson |
> |---|---:|
> | Text Rendering | 0.9657 |
> | Pitch Accuracy | 0.5544 |
> | Facial Consistency | 0.8270 |
> | Speech Intelligibility & Coherence | 0.8300 |
> | Physical Plausibility | 0.8290 |
> | Holistic Semantic Alignment | 0.8402 |
>
> This shows **strong human-model alignment on 5/6 dimensions**. The weaker case, **Pitch Accuracy**, mainly reflects a **floor effect**: current T2AV systems perform extremely poorly on this dimension (best model only **11.25/100**), and human ratings are almost entirely concentrated at **1–2** （maximum 5).
>
> We also conducted an **inter-rater agreement** study on a fixed shared subset of **85 tasks** with all **10 raters**. For pointwise dimensions, the weighted κ is **0.9116** for **Text Rendering** and **0.3156** for **Pitch Accuracy**. For pairwise dimensions, **Cohen’s κ** is **0.8511 / 0.9272 / 0.8455 / 0.8909** for **Facial Consistency / Speech Intelligibility & Coherence / Physical Plausibility / Holistic Semantic Alignment**, respectively. Thus, the validation is **not limited to three dimensions and two raters**. This also addresses **Q5**.
>
> ### W4 \& Q6 Benchmark size and category distribution
> Our benchmark contains **235 prompts across 11 subcategories**, with **20 prompts per category** except **Musical Instrument Tutorial (35)**, which is larger because it involves richer note-level controllability requirements.
>
> ### W5 \& W6 \& Q2 Presentation issues and missing references
> We thank the reviewer for pointing out these presentation issues. We acknowledge the missing figure references, abbreviation inconsistencies, and figure readability problems, and will correct them in the final version. We also note that **Figure 3** is currently missing an explicit in-text reference, and that **“Vis”** in Section 4.2 should be explicitly defined as the **Q-Align visual quality score**.
>
> ### Q1 Different construction pipelines for Ads and Movie Trailers
> **Commercial Ads** were obtained by Gemini-based reverse captioning of YouTube bumper ads; **Movie Trailers** were written from scratch with **GPT-5.2**. Thus, the difference reflects prompt construction rather than evaluation inconsistency.

---

> > ### Author Rebuttal · Reviewer_Go7R · 2026-04-03
> >
> > The rebuttal improves the paper by clarifying that the intended contribution is a task-driven benchmark for fine-grained joint AV semantic evaluation rather than simply longer prompts or more metrics. I also appreciate the added benchmark-size clarification and the expanded human-validation results described in the rebuttal. However, my main concerns are only partially resolved.
> > 1. The novelty claim still needs a more explicit and balanced comparison against the broader recent T2AV benchmark landscape; otherwise the paper still risks reading as an extension of existing benchmark design patterns rather than a clearly new benchmark frontier.
> > 2. The explanation for using different prompt-construction pipelines within the same Professional Media domain is informative but does not fully rule out a construction-style confound.
> >
> > Therefore, while the rebuttal strengthens the submission, it does not fully address the core novelty and validation concerns, so I remain weak accept for now.

---

> > > ### Author Response · Authors · 2026-04-03
> > >
> > > Dear Reviewer Go7R,
> > >
> > > Thank you for your prompt response and for acknowledging the improvements in our rebuttal. We deeply appreciate your continued engagement; your constructive feedback has been instrumental in helping us sharpen how we position this benchmark within the broader evaluation landscape.
> > >
> > > ### 1. Novelty: Pushing the Frontier for Next-Generation T2AV Models
> > >
> > > We completely agree that our novelty must be stated more explicitly and with balanced respect for the existing T2AV benchmark landscape. Our core motivation stems from a critical shift in the application perspective: as state-of-the-art T2AV models (such as Sora 2, Veo 3.1, and Kling 2.6) rapidly advance, they increasingly master coarse audio-visual aesthetics and high-level relevance. The new evaluation frontier—and the real bottleneck for practical, professional application—is fine-grained, complex semantic control.
> > >
> > > Overall, the manuscript examines the central question of how to rigorously quantify these emerging, highly challenging capabilities as models transition from superficial texture generators to physically grounded audio-visual world simulators. While prior benchmarks laid vital groundwork, they target different slices of this evolving problem:
> > >
> > > * **TAVGBench** heavily utilizes distribution-based (e.g., FAD, FVD) and embedding-based metrics (e.g., ImageBind). These are excellent for assessing general fidelity and coarse alignment but fundamentally struggle to capture strict prompt adherence or specific semantic failures (e.g., generating an A4 note vs. a C Major chord) required by real-world users.
> > > * **MTAVG-Bench** provides an essential framework for multi-talker audio-video generation. However, its dialogue-centric scope naturally excludes broader, highly complex daily-life scenarios like general sound design, object interactions, or non-speech physical event generation (such as chemical reactions and acoustic resonance).
> > > * **VABench** and **T2AV-Compass** evaluate broader scenarios and are closer to our setting. However, AVGen-Bench provides a significantly more targeted and challenging set of evaluation dimensions. For instance, while VABench checks general realism, it lacks explicit, isolated quantification for the most failure-prone extremes like strict musical pitch accuracy or multi-shot facial consistency.
> > > * Furthermore, rather than relying predominantly on a direct MLLM judge (as seen in T2AV-Compass), we employ a tighter **specialist small-model parsing + MLLM reasoning** pipeline. For example, our text rendering module uses OCR to extract and aggregate text sequences *before* MLLM reasoning, explicitly preventing long-context overload and ensuring a precise, objective evaluation of legibility.
> > >
> > > Through our framework, The authors appear to assess a significant issue: the pronounced gap between strong audio-visual aesthetics and weak fine-grained semantic reliability. AVGen-Bench explicitly tests the highest number of dimensions in the current literature, introducing unprecedented, challenging tasks (e.g., precise music theory adherence, complex physical simulations). By focusing on these harder, fine-grained tasks, AVGen-Bench is not merely an extension of existing patterns, but a timely and necessary diagnostic tool to guide the next generation of T2AV development.
> > >
> > > ### 2. Prompt-Construction Pipelines in the Professional Media Domain
> > >
> > > You raise a very insightful point regarding the potential for a construction-style confound within the Professional Media domain. We want to clarify that our intention behind using different pipelines (e.g., Gemini-based reverse captioning for Ads versus from-scratch GPT-5.2 generation for Trailers) was strictly to maximize **prompt diversity** and minimize linguistic dataset bias.
> > >
> > > In real-world applications, user prompts are stylistically messy and diverse. Because our evaluation methodology does not make claims or calculate statistical significance *between* the prompt subcategories themselves, this variance does not introduce a confound into our core findings. Every model evaluated in our benchmark faces the exact same fixed, diverse prompt set, ensuring strictly fair model-to-model comparisons across the core evaluation dimensions (Text Rendering, Speech, Physics, etc.).
> > >
> > > We recognize that this could have been explained more clearly. In the final version, we will explicitly clarify the scope and purpose of this prompt diversity, and acknowledge that future benchmark iterations could benefit from exploring a completely unified construction protocol across all subcategories.
> > >
> > > We will carefully revise our manuscript to explicitly claim our contributions along these application-driven lines, ensuring the distinct value of AVGen-Bench is obvious against the current landscape. We hope this enhanced perspective fully resolves your remaining concerns and demonstrates the timely, practical contribution our work brings to the community.

---

### Decision · Program_Chairs · 2026-04-30

**Decision:**

Accept (regular)

**Comment:**

Summary:
This paper proposes AVGen-Bench, a task-driven benchmark for text-to-audio-video generation (T2AVG) across 11 real-world categories. Through experiments, the authors show a strong gap between strong audio-visual aesthetics and weak semantic reliability.

Justifications:
The benchmark of T2AVG is underdeveloped in the research community. The work is useful for evaluating audio-video content generated from text prompts. All reviewers recommended acceptance of the papers (4 - weak accept). The rebuttals addressed most of the reviewers' concerns.